# Mean ocean temperature change and decomposition of the benthic $\delta^{18}$O record over the past 4.5 million years

**Peter U. Clark**[1,2,⚲,★], **Jeremy D. Shakun**[3,★], **Yair Rosenthal**[4,5], **Chenyu Zhu**[6], **Patrick J. Bartlein**[7], **Jonathan M. Gregory**[8,9], **Peter Köhler**[10], **Zhengyu Liu**[11], and **Daniel P. Schrag**[12]

[1]College of Earth, Ocean, and Atmospheric Sciences, Oregon State University, Corvallis, OR 97331, USA

[2]School of Geography and Environmental Sciences, University of Ulster, Coleraine, BT52 1SA, Northern Ireland, UK

[3]Department of Earth and Environmental Sciences, Boston College, Chestnut Hill, MA 02467, USA

[4]Department of Marine and Coastal Science, Rutgers, The State University of New Jersey, New Brunswick, NJ 08901, USA

[5]Department of Earth and Planetary Sciences, Rutgers, The State University of New Jersey, New Brunswick, NJ 08901, USA

[6]Institute of Atmospheric Physics, Chinese Academy of Sciences, Beijing, China

[7]Department of Geography, University of Oregon, Eugene, OR 97403-1251, USA

[8]National Center for Atmospheric Science, University of Reading, Reading, UK

[9]Met Office Hadley Centre, Exeter, UK

[10]Alfred-Wegener-Institut Helmholtz-Zentrum für Polar- und Meeresforschung, Bremerhaven, Germany

[11]Department of Geography, The Ohio State University, Columbus, OH 43210, USA

[12]Department of Earth and Planetary Sciences, Harvard University, Cambridge, MA 02138, USA

⚲*Invited contribution by Peter U. Clark, recipient of the EGU Milutin Milanković Medal 2024.*

★These authors contributed equally to this work.

**Correspondence:** Peter U. Clark (clarkp@onid.orst.edu)

**Abstract.** We use a recent reconstruction of global mean sea surface temperature change relative to preindustrial ($\Delta$GMSST) over the last 4.5 Myr together with independent proxy-based reconstructions of bottom water ($\Delta$BWT) or deep-ocean ($\Delta$DOT) temperatures to infer changes in mean ocean temperature ($\Delta$MOT). Three independent lines of evidence show that the ratio of $\Delta$MOT / $\Delta$GMSST, which is a measure of ocean heat storage efficiency (HSE), increased from $\sim 0.5$ to $\sim 1$ during the Middle Pleistocene Transition (MPT, 1.5–0.9 Ma), indicating an increase in ocean heat uptake (OHU) at this time. The first line of evidence comes from global climate models; the second from proxy-based reconstructions of $\Delta$BWT, $\Delta$MOT, and $\Delta$GMSST; and the third from decomposing a global mean benthic $\delta^{18}$O stack ($\delta^{18}O_b$) into its temperature ($\delta^{18}O_T$) and seawater ($\delta^{18}O_{sw}$) components. Regarding the latter, we also find that further corrections in benthic $\delta^{18}$O, probably due to some combination of a long-term diagenetic overprint and to the carbonate ion effect, are necessary to explain reconstructed Pliocene sea-level highstands inferred from $\delta^{18}O_{sw}$. We develop a simple conceptual model that invokes an increase in OHU and HSE during the MPT in response to changes in deep-ocean circulation driven largely by surface forcing of the Southern Ocean. Our model accounts for heat uptake and temperature in the non-polar upper ocean (0–2000 m) that is mainly due to wind-driven ventilation, while changes in the deeper ocean ($> 2000$ m) in both polar and non-polar waters occur due to high-latitude deepwater formation. We propose that deepwater formation was substantially reduced prior to the MPT, effectively decreasing HSE. We attribute these changes in deepwater formation across the MPT to long-term cooling which caused a change starting $\sim 1.5$ Ma from a highly stratified Southern Ocean due to warm SSTs and reduced sea-ice extent to a Southern Ocean which, due to colder SSTs and increased sea-ice extent, had a greater vertical exchange of water masses.

## 1   Introduction

The ocean is one of the largest heat reservoirs in the climate system. Changes in ocean heat content play an important role in mitigating the Earth's surface temperature response to a radiative forcing (Gregory et al., 2002) and also influence sea-level change (Church et al., 2013) and ocean stratification, with the latter affecting the rate of ocean heat uptake (OHU) (Newsom et al., 2023), the efficiency of the oceanic carbon sink (Bronselaer and Zanna, 2020), and large-scale circulation (Fox-Kemper et al., 2021). Proxy-based reconstructions of changes in bottom water temperature ($\Delta$BWT), deep-ocean temperature ($\Delta$DOT), and mean ocean temperature ($\Delta$MOT)[1] relative to preindustrial (PI) identify large variations in ocean temperature, ocean heat content, and energy imbalance on $10^3$–$10^6$-year timescales (Baggenstos et al., 2019; Rohling et al., 2022; Shackleton et al., 2023). These include orbital-scale variations of $\sim$ 2.5 to 3.5 °C over the last $\sim$ 0.7 Myr (Sosdian and Rosenthal, 2009; Elderfield et al., 2012; Shakun et al., 2015; Haeberli et al., 2021; Shackleton et al., 2023; Martin et al., 2002) and a long-term cooling over much of the past 4.5 Myr (Bates et al., 2014; Hansen et al., 2013; Lear et al., 2003; Rohling et al., 2021; Cramer et al., 2011; de Boer et al., 2014; Rohling et al., 2022; Westerhold et al., 2020; Evans et al., 2024) (Fig. 1).

In detail, however, these long-term reconstructions can differ by as much as 2–3 °C (Fig. 1). Differences among the reconstructions on these longer timescales may reflect some combination of (1) differences in the calibration of Mg / Ca-based reconstructions as well as in the process of accounting for changes in its seawater (sw) ratio Mg / Ca$_{sw}$ and carbonate ion concentration (Cramer et al., 2011; Rosenthal et al., 2022), (2) proxy-based reconstructions that sample local BWT that is not representative of DOT or MOT (Lear et al., 2003; Woodard et al., 2014), (3) an unaccounted-for

---

[1]The terms "bottom water temperature", "deep-ocean temperature", and "deep-sea temperature" are commonly used interchangeably in paleoceanography to refer to depths $> 200$ m. In some cases, these terms have been used for the temperature of a specific site (Waelbroeck et al., 2002; Sosdian and Rosenthal, 2009; Elderfield et al., 2012) or for the entire ocean $> 200$ m (Hansen et al., 2013; Rohling et al., 2022). The latter meaning is not equivalent to "mean ocean temperature" (MOT) since that includes the surface ocean, but the small volume of this surface layer means that the global ocean temperature $> 200$ m will be comparable to MOT. Finally, these terms do not distinguish between water depths that are commonly used to describe processes and patterns of ocean heat storage: sea surface (skin) with zero heat capacity, upper (0–700 m), intermediate (700–2000 m), deep (2000–4000 m), and abyssal ($> 4000$ m) (Purkey et al., 2019; Fox-Kemper et al., 2021; Cheng et al., 2022). Here we use the term "bottom water temperature" (BWT) for a site-specific temperature reconstruction and "deep-ocean temperature" (DOT) for temperature of the whole ocean that is $> 200$ m. We also distinguish among the depth layers (upper, intermediate, deep, and abyssal) when discussing ocean heat storage.

nonstationarity in the relationship between $\delta^{18}O_b$ and sea level that is used to derive BWT or DOT (Bates et al., 2014; Rohling et al., 2021; Rohling et al., 2022; Waelbroeck et al., 2002), and (4) scaling of a $\delta^{18}O_b$ range to an inferred DOT range (LGM–Holocene) (Hansen et al., 2013; Westerhold et al., 2020; Hansen et al., 2023) that assumes stationarity in this scaling over the last 4.5 Ma and underestimates the DOT range by $\sim 70$ % (Haeberli et al., 2021; Shackleton et al., 2023). It is thus important to narrow these uncertainties in the ocean's temperature history in order to gain a better understanding of the role of the ocean heat reservoir in the Earth's energy budget as well as the processes that contribute to OHU.

Here we find that climate models and proxy data from the last 4.5 Myr show good agreement in the ratio between $\Delta$MOT and changes in global mean sea surface temperature ($\Delta$GMSST), which Zhu et al. (2024) defined as the ocean heat storage efficiency (HSE $= \Delta$MOT / $\Delta$GMSST) when in equilibrium, suggesting that we can use a new $\Delta$GMSST reconstruction (Clark et al., 2024) to derive $\Delta$MOT over this interval.[2] A particularly notable finding is that HSE increased from $\sim 0.5$ to $\sim 1$ when $\Delta$GMSSTs decreased to below $\sim 0$ °C, corresponding to the start of the Middle Pleistocene Transition (MPT, 0.9–1.5 Ma). We use this understanding of $\Delta$MOT to decompose the benthic $\delta^{18}O$ record ($\delta^{18}O_b$) into its temperature ($\delta^{18}O_T$) and seawater ($\delta^{18}O_{sw}$) components, with this decomposition providing independent support for the need to increase HSE during the MPT. We then summarize the processes that contribute to changes in ocean heat storage and mean ocean temperature which in turn provide the basis for a simple conceptual model that explains why HSE may have increased during the MPT.

## 2   Derivation of ocean heat storage efficiency

### 2.1   Constraints from climate models

We obtained $\Delta$MOTs and $\Delta$GMSSTs from 93 coupled ocean–atmosphere climate model equilibrium simulations that were run with atmospheric $CO_2$ concentrations ranging from 180 ppm to 9 times PI (Fig. 2) (Alder and Hostetler, 2015; Galbraith and de Lavergne, 2019; Haywood et al., 2020; Clark et al., 2016; Goudsmit-Harzevoort et al., 2023; Rugenstein et al., 2019; He, 2011; Bereiter et al., 2018). Here we use four regression models to describe the relationship

---

[2]HSE can be interpreted as proportional to the effective heat capacity of the ocean for equilibrium changes, while the change in ocean heat content (in $J$) can be accounted for by regarding it as the (SST) $\times$ (HSE) $\times$ (the heat capacity of the ocean). Because HSE refers to equilibrium, it differs from ocean heat uptake efficiency, which specifically describes transient climate states on decadal timescales during which the non-equilibrium of the ocean that causes a substantial rate of ocean heat uptake (in W m$^{-2}$) is large enough to affect the surface climate through its effect on the energy balance.

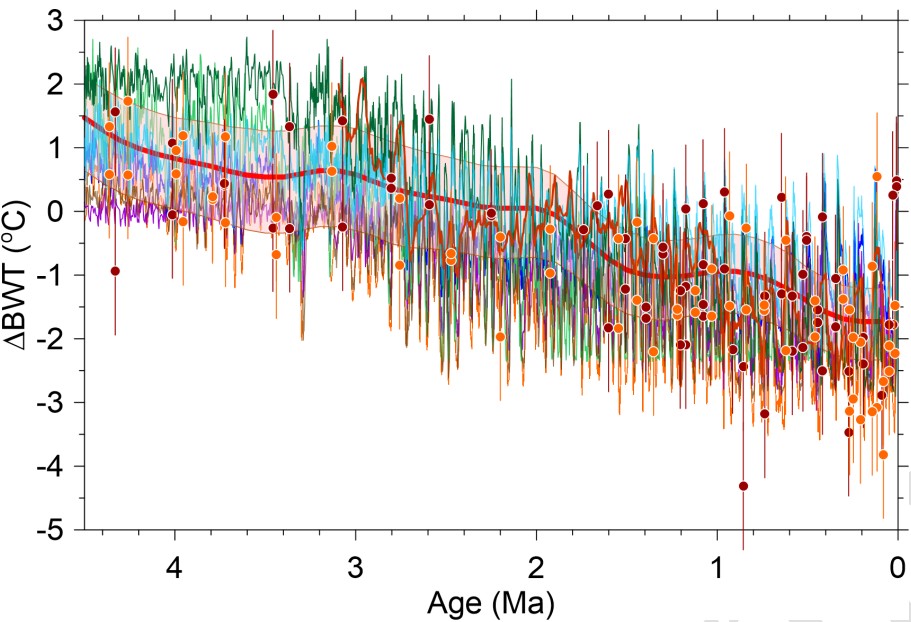

**Figure 1.** Reconstructions of changes in bottom water temperature ($\Delta$BWT) since 4.5 Ma. Dark-green line is from Evans et al. (2024), referenced to preindustrial (PI). Purple line is from Rohling et al. (2021), referenced to PI. Dark-brown line is from Rohling et al. (2022), referenced to PI. BWTs were mean-shifted by 2.1 °C so that they have the same long-term mean as our $\Delta$MOT reconstruction over the last 400 kyr. Blue line is the $\Delta$BWT reconstruction from Hansen et al. (2013). BWTs were mean-shifted by 1 °C so that they are referenced to the Holocene (i.e., 0 °C for the last 10 kyr). Light-green line is the $\Delta$BWT reconstruction from Bates et al. (2014). BWTs were mean-shifted by 1.46 °C so that they are referenced to the Holocene (i.e., 0 °C for the last 10 kyr). Orange line is the $\Delta$MOT reconstruction from de Boer et al. (2014), referenced to PI. Orange symbols with $1\sigma$ uncertainty are the $\Delta$BWT reconstruction from ODP site 806 (Lear et al., 2003). Mg / Ca BWTs from this data set were mean-shifted by $-1.73$ °C CE1 to fall within same range as our $\Delta$MOT reconstruction for the last 800 kyr CE2. Red-brown symbols with $1\sigma$ uncertainty are the $\Delta$BWT reconstruction from ODP site 926 (Lear et al., 2003). Mg / Ca BWTs from this data set were mean-shifted by $-1.73$ °C CE3 so that they are referenced to the Holocene (i.e., 0 °C for the last 10 kyr). Red bold line with uncertainty in pink (90 % confidence interval) is the smoothed Mg / Ca-based $\Delta$BWT reconstruction from Cramer et al. (2011) removing periods shorter than 2 Myr and their Eq. (7b) for Mg / Ca temperature calibration. TS3

between $\Delta$MOT and $\Delta$GMSST in this synthesis of climate model output. The four models were fit using R *stats* package functions *lm*, *segmented*, *loess*, and *smoothing.spline*, as well as their associated functions for prediction and diagnostic analyses (R, 2024) (see Sect. S1 in Supplement). The relationship between $\Delta$MOT and $\Delta$GMSST is broadly linear, with a slope and intercept close to 1.0 and 0.0, respectively (Figs. 2a, S1A); a high $R^2$ value (0.924); a highly significant $F$-test statistic (1104, $p < 0.000$ TS4); and a relatively low residual standard error (SE; 1.184 TS5). However, a residual diagnostic analysis of this model reveals a subtle but significant nonlinear lack of fit which can be seen in the residual scatter diagram (Fig. S1B). The nonlinearity in the residual scatter diagram is largely related to a quasi-linear cluster of points that plot below the main cloud of points (Fig. 2a), which include the points with the 10-largest negative residuals. The climate models that produced these data points are generally a mix of higher-than-present $CO_2$ simulations and sensitivity tests, but not all such experiments produce that pattern of large negative residuals. This eliminates the possibility that there is an underlying family-of-curves explanation for the apparent nonlinearity. The residuals are

also not normally distributed (Shapiro–Wilk test $W = 0.932$, $p < 0.001$) and in fact are visibly bimodal, imparted by the cluster of negative residuals.

Inspection of the scatter diagrams suggests that an alternative to an overall straight-line fit might be a segmented linear fit (Muggeo, 2003), with the individual segments approximately defined by values of $\Delta$GMSST less than 1.0 °C, between 1.0 and 5.0 or 6.0 °C, and greater than 6.0 °C. To test that notion, we used the R *segmented* function. The best fit (in terms of goodness-of-fit and residual diagnostics) was a model with two breakpoints, at 1.187 (SE = 1.307 TS6) and 5.290 (SE = 0.855), and slopes (i.e., HSE) for the lower, middle, and upper segments of 1.000 (SE = 0.178), 0.575 (SE = 0.185), and 1.244 (SE = 0.066), respectively (Figs. 2b, S1C). The model fit is slightly better than that for the linear one, with an adjusted $R^2$ value of 0.939 (vs. 0.924 for the linear one). The segmented regression thus implies that there are distinct $\Delta$GMSST thresholds (the breakpoints) where HSE changes.

We next considered models that can accommodate smooth variations in slope and can be fit via local- or nonparametric-regression approaches. Of these, we first considered a fit us-

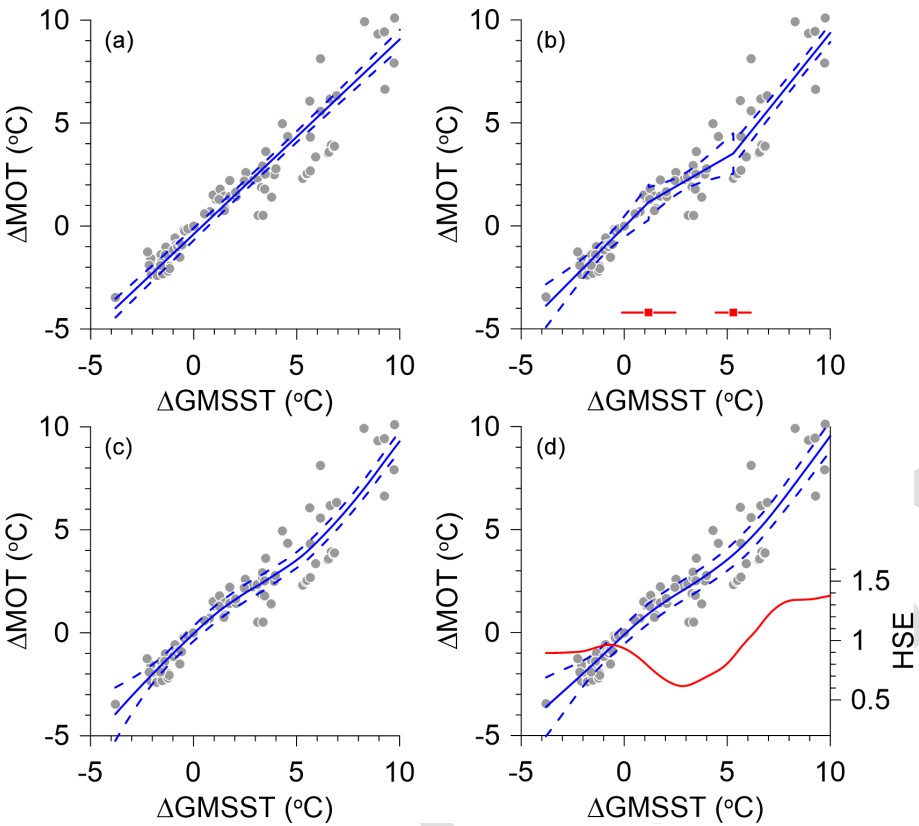

**Figure 2.** Relationship between equilibrium changes in mean ocean temperature ($\Delta$MOT) and global mean sea surface temperature ($\Delta$GMSST) from preindustrial (PI) established from 93 simulations by climate models. Results shown are from PMIP3 model runs for LGM boundary conditions (Braconnot et al., 2012), model runs for $CO_2 \leq 270$ ppm with no ice sheets and with LGM ice sheets (Galbraith and de Lavergne, 2019), model runs for the last deglaciation (Alder and Hostetler, 2015), model runs for $CO_2 \geq 420$ ppm (Clark et al., 2016), LongRunMIP model runs for $CO_2 \geq 2\times$ PI (Rugenstein et al., 2019), model runs for $CO_2$ ranging from $1\times$ to $9\times$ PI (Goudsmit-Harzevoort et al., 2023), PlioMIP2 model runs with $CO_2 = 400$ ppm (Haywood et al., 2020), and model runs for $CO_2 \geq 405$ ppm (Galbraith and de Lavergne, 2019). Fits use four different regression models: **(a)** *lm*, **(b)** *segmented*, **(c)** *loess*, and **(d)** *smoothing.spline*. The red dots and lines in panel **(b)** show the estimated breakpoints in the segmented regression (dots) and their uncertainties. The red line in panel **(d)** shows the calculated first derivative of the fitted values, i.e., the heat storage efficiency (HSE = $\Delta$MOT / $\Delta$GMSST).

ing the R *loess* function (Cleveland et al., 1992). The fitted values (Fig. 2c, S1E) were broadly similar to the individual slopes in the segmented regression (Fig. 2b), but the slope varied smoothly instead of abruptly at $\Delta$GMSST values around 1.0 and 6.0 °C. This model fits better than the two previous ones (AIC = 277.13; adjusted $R^2 = 0.940$) (Fig. S1F). Finally, we considered a fit using a smoothing spline (Hastie, 1992). Among the three alternatives to the linear model, this model has several desirable properties – it fits the data best, with the lowest AIC and highest adjusted $R^2$ (although the differences are quite small) (Fig. S1G), and is quite smooth while still displaying the changes in slope explicitly represented by the segmented regression model (Fig. 2d). In addition, its associated *predict* function can be used to extract the local numeric first derivative of the fitted curve, which provides a representation of the HSE. This shows that HSE is $\sim 1$ when $\Delta$GMSST is $< 0$ °C and different from 1 when $\Delta$GMSST is between $\sim 0$ and 6 °C (red curve in Figs. 2d

and S1G), again consistent with the $\Delta$GMSST thresholds identified by the segmented and loess functions. Given these properties, the smoothing spline regression is our preferred choice for representing the relationship between $\Delta$MOT and $\Delta$GMSST in our synthesis of climate model results.

## 2.2 Constraints from proxy reconstructions

We next derive HSE from proxy reconstructions of $\Delta$MOT, $\Delta$BWT, and $\Delta$GMSST. $\Delta$MOT reconstructions spanning the past 0.7 Myr are based on Antarctic ice-core noble gases. We update Mg / Ca-based $\Delta$BWT reconstructions with newer temperature calibrations (where necessary) and account for variations in the seawater Mg / Ca concentration ratio for long-term temperature reconstructions ($> 1$ Ma) that extend into the Early Pleistocene and the Pliocene (see Methods in Supplement). Lastly, we use a $\Delta$GMSST reconstruction for the past 4.5 Myr that has negligible loss of variability from

stacking of individual SST records with differing resolutions and age-model uncertainties (Clark et al., 2024). This reconstruction suggests that average (401 kyr) $\Delta$GMSSTs decreased from $\sim 3\,°\mathrm{C}$ at 4 Ma to $0\,°\mathrm{C}$ at 1.5 Ma, falling in the range that climate models suggest should correspond to an HSE smaller than 1 (Figs. 2d, S1). Average cooling then accelerated between 1.5–0.9 Ma, corresponding to the MPT (Clark et al., 2024). $\Delta$GMSSTs then remained largely ($> 95\,\%$ of the time) below $0\,°\mathrm{C}$ over the remainder of the Pleistocene, falling in the range that climate models suggest should correspond to an HSE of $\sim 1$. In the following, we use the $0$–$1\,°\mathrm{C}$ $\Delta$GMSST threshold suggested from climate models to derive proxy-based HSE prior to and since the MPT.

### 2.2.1 Proxy-based HSE prior to the MPT

Figure 3 shows that most $\Delta$BWT and $\Delta$DOT reconstructions covering the interval from 1.5–4.5 Ma fall below $\Delta$GMSSTs, thus identifying an HSE < 1. Differences among the $\Delta$BWT and $\Delta$DOT reconstructions, however, prevent a more accurate derivation of HSE needed to assess climate model predictions and decompose the $\delta^{18}O_b$ record into its $\delta^{18}O_T$ and $\delta^{18}O_{sw}$ components. To first order, these differences reflect the different approaches used for the reconstructions. For example, Waelbroeck et al. (2002) developed an approach that regresses independently known sea-level data (e.g., from corals) on $\delta^{18}O_b$ for the last glacial cycle and then used this regression to reconstruct sea level from $\delta^{18}O_b$ records for the last four glacial cycles. Sea level is converted to $\delta^{18}O_{sw}$ using a constant relation of $\Delta\delta^{18}O_{sw} : \Delta$GMSL. $\delta^{18}O_{sw}$ is then subtracted from the $\delta^{18}O$ record to derive $\delta^{18}O_T$, which is converted to $\Delta$BWT using a constant relation of $\Delta\delta^{18}O_T : \Delta T$ (commonly $0.25\,‰\,°\mathrm{C}^{-1}$). Following this same approach, Siddall et al. (2010) and Bates et al. (2014) extended the regression over the last two glacial cycles to reconstruct $\Delta$DOT from $\delta^{18}O_b$ records over the last 5 Myr (see Fig. 3c for the past 4.5 Myr). Rohling et al. (2021, 2022) further extended the regression over the last 800 kyr using the LR04 $\delta^{18}O_b$ stack and a stack of sea-level records from Spratt and Lisiecki (2016) to reconstruct $\Delta$DOT over the past 40 Myr (see Fig. 3a for the past 4.5 Myr). Rohling et al. (2021, 2022) also accounted for $\delta^{18}O$ variations in land ice ($\delta^{18}O_i$) over the last glacial cycle which were then applied to their sea-level record to derive $\delta^{18}O_{sw}$ from $\delta^{18}O_b$, but this did not include the effect of increasing temperatures on $\delta^{18}O_i$ prior to 0.8 Ma. In any event, applying this regression approach to derive $\delta^{18}O_{sw}$ will, by default, reproduce the variability of the $\delta^{18}O_b$ record, including the increase in the size of glaciations during the MPT, and thus potentially bias the $\Delta$DOT reconstruction.

Another approach scales $\delta^{18}O_b$ to an inferred DOT LGM–Holocene range of $-2\,°\mathrm{C}$ (Hansen et al., 2013; Westerhold et al., 2020; Hansen et al., 2023) and assumes stationarity in this scaling over the last 4.5 Ma (Fig. 3b). However, the ice-core noble-gas estimates suggest that the prescribed LGM cooling of $-2°$ underestimates the MOT range by $\sim 70\,\%$ (Haeberli et al., 2021; Shackleton et al., 2023), and this scaling is likely nonstationary.

Given these and other uncertainties and differing assumptions used to derive $\Delta$DOT from $\delta^{18}O_b$ records (Bates et al., 2014; Rohling et al., 2021; Hansen et al., 2013; Westerhold et al., 2020; Rohling et al., 2022; Hansen et al., 2023; Evans et al., 2024), we derive changes in HSE using Mg / Ca-based $\Delta$BWT reconstructions that extend into the Early Pleistocene and the Pliocene (see Sect. S2 in Supplement). These include two low-resolution $\Delta$BWT reconstructions, one from the equatorial Pacific (ODP site 806, 2500 m) (Fig. 3e) that samples Pacific Deep Water and one from the equatorial Atlantic (ODP site 926, 3500 m) that today is in the mixing zone between North Atlantic Deep Water (NADW) and Antarctic Bottom Water (AABW) (Lear et al., 2003) (Fig. 3f). We also use high-resolution Mg / Ca-based $\Delta$BWT reconstructions from DSDP site 607 (3427 m) (Dwyer and Chandler, 2009; Sosdian and Rosenthal, 2009) and nearby IODP U1313 (3426 m) (Jakob et al., 2020) in the North Atlantic (Fig. 3i). Lastly, we use a smoothed BWT reconstruction based on a compilation of Mg / Ca records derived from six species or genera of benthic foraminifera (Cramer et al., 2011) (Fig. 3h). Although the smoothed reconstruction is based on Mg / Ca data from eight different sites, it is largely based on data from sites 806 and 926 for the last 4.5 Myr.

As discussed further in Sect. 4, temperature change of the ocean interior in response to surface forcing is nonuniform, with warming of the upper half of the ocean being stronger than GMSST warming as opposed to weaker in the lower half of the ocean (Bronselaer and Zanna, 2020; Fox-Kemper et al., 2021; Rugenstein et al., 2016; Zhu et al., 2024). To assess the relationship of $\Delta$BWT at the Pacific and Atlantic core sites to $\Delta$MOT, we use results from the Transient Climate Evolution (iTRACE) simulation performed with the isotope-enabled Community Earth System Model version 1.3 that has reproduced the observed evolution of global climate and water masses from 21 ka to the Early Holocene (11 ka) (Gu et al., 2020; He et al., 2021). The simulation is forced by changes in solar insolation from orbital changes (CE4 ORBs), greenhouse gases (GHGs), reconstructed ice sheets (ICEs), and meltwater fluxes (MWFs), with the latter causing millennial-scale variations in ocean heat storage and MOT through its effect on the Atlantic Meridional Overturning Circulation (AMOC) (Zhu et al., 2024).

Given that the Pacific Ocean constitutes $\sim 50\,\%$ of the total ocean volume, Lear et al. (2003) considered site 806 BWT to closely represent MOT. This inference is consistent with iTRACE $\Delta$BWTs at the location of site 806 that closely parallel $\Delta$MOT throughout the simulation (Fig. 4a). In this regard, we note that the Cramer et al. (2011) reconstruction, being largely weighted by site 806 data for the 1–4.5 Ma interval but with a different calibration than used in Lear et al. (2003), also represents a close approximation of $\Delta$MOT.

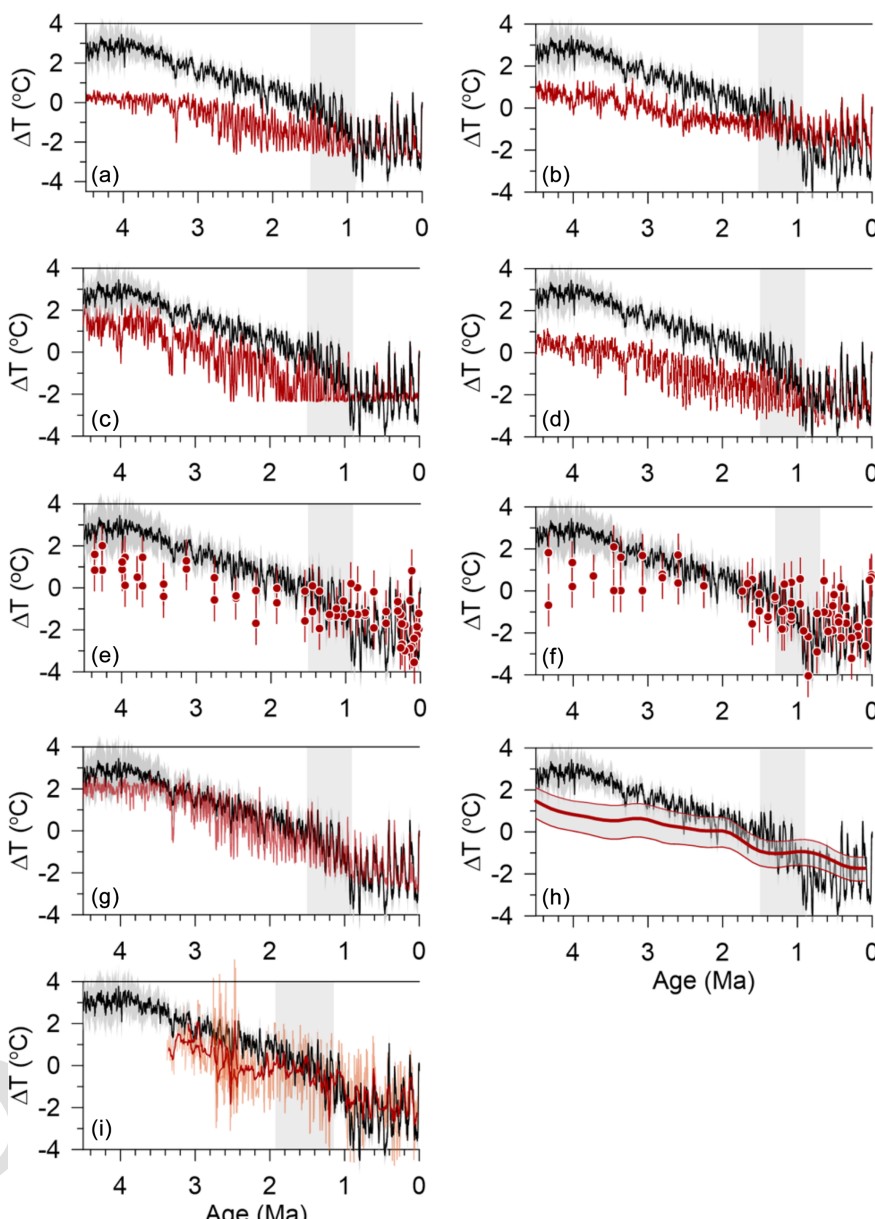

**Figure 3.** The reconstruction of global mean sea surface temperature change from preindustrial (Clark et al., 2024) (black with $1\sigma$ uncertainty in gray) compared to reconstructions of changes in bottom water temperature ($\Delta$BWT, dark red). The vertical gray box represents the Middle Pleistocene Transition (0.9–1.5 Ma). **(a)** $\Delta$BWT reconstruction from Rohling et al. (2022), referenced to preindustrial (PI). **(b)** $\Delta$BWT reconstruction from Hansen et al. (2013). BWTs were mean-shifted by $1\,°C$ so that they are referenced to the Holocene (i.e., $0\,°C$ for the last 10 kyr). **(c)** $\Delta$BWT reconstruction from Bates et al. (2014). BWTs were mean-shifted by $1.46\,°C$ so that they are referenced to the Holocene (i.e., $0\,°C$ for the last 10 kyr). **(d)** $\Delta$MOT reconstruction from de Boer et al. (2014), referenced to PI. **(e)** Mg / Ca $\Delta$BWT reconstruction from ODP site 806 (Lear et al., 2003). Mg / Ca BWTs were mean-shifted by $-1.73\,°C$ to fall within same range as our $\Delta$MOT reconstruction for the last 800 kyr. **(f)** Mg / Ca $\Delta$BWT reconstruction from ODP site 926 (Lear et al., 2003). Mg / Ca BWTs were mean-shifted by $-1.73\,°C$ so that they are referenced to the Holocene (i.e., $0\,°C$ for the last 10 kyr). **(g)** Deep-ocean temperature reconstruction from Evans et al. (2024), which is the Rohling et al. (2022) reconstruction **(a)** with pH-corrected benthic $\delta^{18}$O, referenced to PI. **(h)** The 2 Myr smoothed $\Delta$BWT reconstruction from Cramer et al. (2011) using their Eq. (7b) with 90 % confidence interval. BWTs were mean-shifted by $2.1\,°C$ so that they have the same long-term mean as our $\Delta$MOT reconstruction over the last 400 kyr. **(i)** Mg / Ca $\Delta$BWT from North Atlantic DSDP site 607 for $> 2.9$ Ma (Dwyer and Chandler, 2009) and $< 2.8$ Ma (Sosdian and Rosenthal, 2009) (11-point running average in dark red) and $\Delta$BWT from North Atlantic IODP site U1313 for 2.4–2.75 Ma (Jakob et al., 2020). Site 607 data are mean-shifted by $-1.73\,°C$ so that they are referenced to the Early Holocene (i.e., $0\,°C$ at 10 ka). Site U1313 data are referenced to modern BWT derived by Mg / Ca measurements on core-top samples.

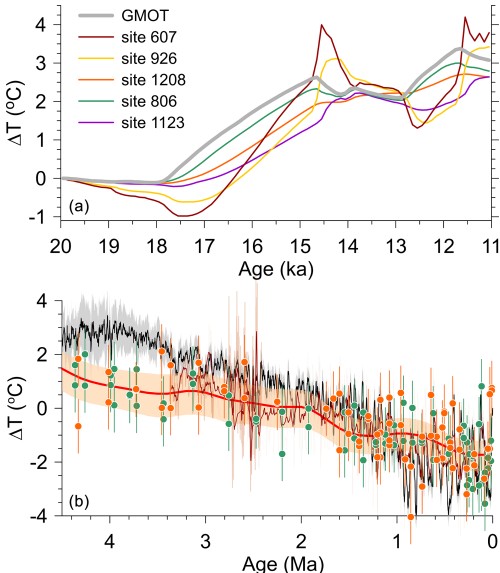

**Figure 4. (a)** Simulated changes in global mean ocean temperature (MOT) relative to 20 ka from the iTRACE experiment as well as for five sites with $\Delta$BWT reconstructions (Zhu et al., 2024). **(b)** Global mean sea surface temperature change (black line, $1\sigma$ uncertainty) (Clark et al., 2024) compared to Mg / Ca-based $\Delta$BWT reconstructions from Pacific ODP site 806 (green circles) (Lear et al., 2003), North Atlantic ODP site 926 (orange circles) (Lear et al., 2003), North Atlantic site 607 for > 2.9 Ma (Dwyer and Chandler, 2009) and < 2.8 Ma (Sosdian and Rosenthal, 2009) and site U1313 for 2.4–2.75 Ma (Jakob et al., 2020) (9-, 11-, and 21-point running averages, respectively, in dark red), and smoothed reconstruction from Cramer et al. (2011) using their Eq. (7b) with 90 % confidence interval.

On the other hand, sites 926, 607, and U1313 were largely bathed by NADW prior to $\sim 1.5$ Ma, after which NADW shoaled and was replaced by AABW at these sites during glaciations (Lisiecki, 2014). In both cases, the associated water masses account for a significantly smaller fraction of total ocean volume, and their BWTs may thus not reflect MOT. Despite being under a greater influence of millennial-scale changes in the AMOC, simulated $\Delta$BWTs at Atlantic sites 926 and 607 warm as much as MOT by 11 ka (Fig. 4a). These similar changes in $\Delta$BWT are further shown when combining the $\Delta$BWT data from sites 806 and 926 (Lear et al., 2003), the smoothed $\Delta$BWT reconstruction from Cramer et al. (2011) (comprised largely of data from sites 806 and 926), and the $\Delta$BWT data from sites 607 (Dwyer and Chandler, 2009; Sosdian and Rosenthal, 2009) and U1313 (Jakob et al., 2020) (Fig. 5b). These results thus suggest that $\Delta$BWTs at the Pacific and Atlantic sites agree and are closely monitoring $\Delta$MOT, suggesting that we can combine their $\Delta$BWT reconstructions to approximate $\Delta$MOT.

We assess which HSE best agrees with the Mg / Ca $\Delta$BWT data before 1.5 Ma by comparing the combined data from sites 806, 926, and 607 (Fig. 5a–c) and the smoothed

reconstruction from Cramer et al. (2011) (based on sites 806 and 926 with a different calibration) (Fig. 5d–f) to three different scenarios of $\Delta$MOT derived from our $\Delta$GMSST reconstruction based on HSEs of 1, 0.5, and 0.1. The uncertainty on the $\Delta$MOT-scenario reconstructions is calculated as the square root of the sum of squares of the uncertainty in the $\Delta$GMSST reconstruction and in the HSE derived from models (Fig. 2). We attribute the total spread in the Mg / Ca data around their long-term trend to some combination of climate variability, analytical and calibration errors, and site location.

Based on their similar slopes, we find that the best agreement with the Mg / Ca-based $\Delta$BWT data is for an HSE of $\sim 0.5$, with long-term rates of cooling between 1.5–4.5 Ma in both the data and our corresponding $\Delta$MOT reconstruction being 0.6–0.8 °C Myr$^{-1}$ (Figs. 5b, e, 6). HSE values of 0.7 and 0.3 closely encompass the uncertainty on our $\Delta$MOT reconstruction based on an HSE $= 0.5$ (Figs. 5b, e, S2A), suggesting an HSE of $0.5 \pm 0.2$. Any further decrease in HSE would suggest virtually no change in MOT between 1.5–4.5 Ma, in contrast to the long-term cooling trend shown by all $\Delta$BWT and $\Delta$DOT reconstructions (Figs. 1, 3, 5b, e, 6).

### 2.2.2 Proxy-based HSE since the MPT

The low-resolution Antarctic ice-core noble-gas proxy records of $\Delta$MOT for the past 0.7 Myr (Haeberli et al., 2021; Shackleton et al., 2023, 2020, 2021; Bereiter et al., 2018) provide the most robust measure for deriving HSE. Over this interval, there is good agreement ($R^2 = 0.75$) between the ice-core records of $\Delta$MOT and $\Delta$GMSST with an HSE of $\sim 0.9$, which, within uncertainties, agrees with an HSE of $\sim 1$ found in climate models for $\Delta$GMSSTs < 0 °C (Fig. 7a, b, c). The temporal and amplitude changes in $\Delta$BWT and $\Delta$GMSST reconstructions extending back to 0.8 Ma also largely agree (Fig. 7e–k), with dominant orbital-scale variability. We note one exception in the ODP site 1123 reconstruction (Elderfield et al., 2012), where glacial $\Delta$BWTs tend to be warmer than $\Delta$GMSSTs over the past 0.8 Myr (Fig. 7d). We attribute these differences to site 1123 capturing a regional signal of glacial $\Delta$BWT that differs from $\Delta$MOT, although calibration uncertainties for this Mg / Ca record and changes in carbonate saturation, which are greatest at low temperatures, may also play a role. Otherwise, the general agreement between the other $\Delta$BWT reconstructions and the $\Delta$GMSST reconstruction further supports an HSE of $\sim 1$ over the last 0.8 Myr when $\Delta$GMSSTs were < 0 °C, consistent with model results.

Multiple reconstructions of $\Delta$GMSST and $\Delta$MOT that span the Last Glacial Maximum (LGM, 26–18 ka) provide further constraints on HSE at that time (Fig. 7l). There is good agreement among $\Delta$GMSST reconstructions (referenced to PI), with proxy-based estimates of $-3.0 \pm 0.1$ °C (18–22 ka average) (Shakun et al., 2012), $-2.9$ °C ($-3.0$ to $-2.7$ °C, 95 % confidence interval (CI) (23–19 ka average) (Tierney et al., 2020), and $-3.3 \pm 0.4$ °C (26–18 ka av-

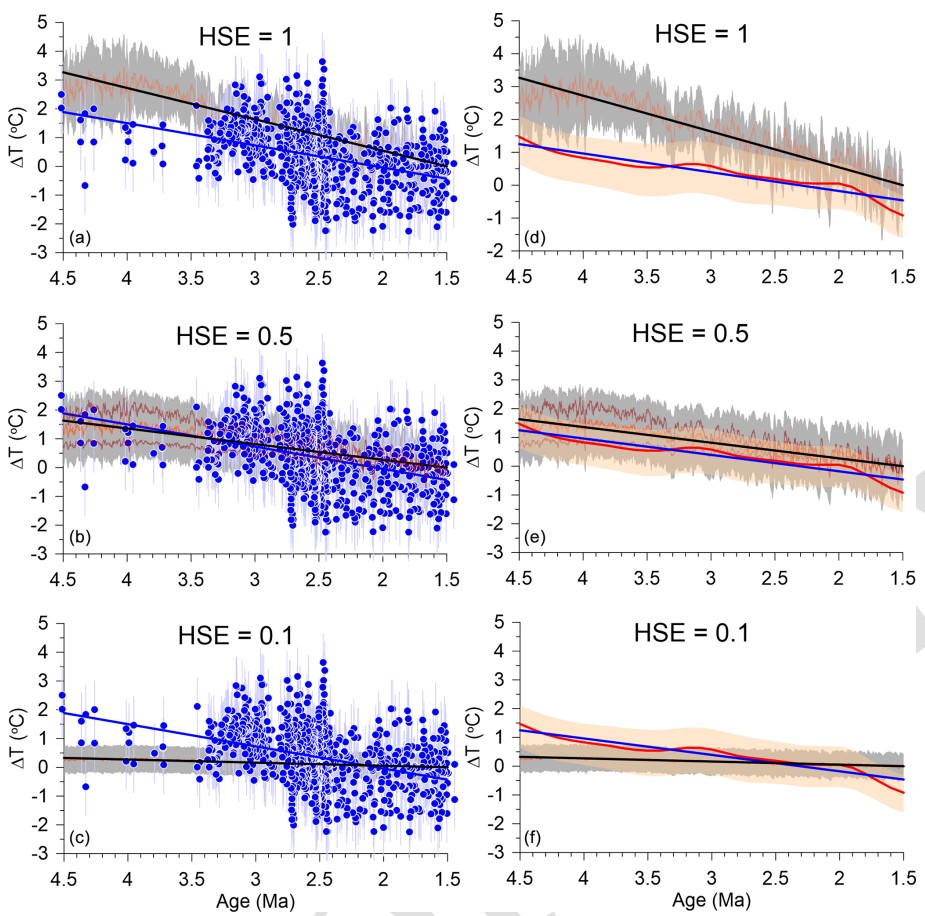

**Figure 5. (a–c)** Our reconstructed global mean ocean temperature anomaly ($\Delta$MOT) for 1.5–4.5 Ma based on three different HSEs (orange lines) compared to proxy-based (Mg / Ca) reconstructions of bottom water temperature anomalies (blue) (1 °C uncertainty) (Sosdian and Rosenthal, 2009; Lear et al., 2003). The $1\sigma$ uncertainty on $\Delta$MOT reconstructions (gray fill) derived by combining in quadrature $1\sigma$ uncertainties on our $\Delta$SST reconstruction and on the relationship between $\Delta$GMSST and $\Delta$MOT (Fig. 2a). Regression lines plotted through $\Delta$MOT reconstructions (black) and Mg / Ca data (blue). **(a)** Our reconstructed $\Delta$MOT based on HSE = 1. **(b)** Our reconstructed $\Delta$MOT based on HSE = 0.5. Also shown are $\Delta$MOT reconstructions based on HSE = 0.7 and HSE = 0.3 (brown lines), which closely encompass the $1\sigma$ uncertainty on our $\Delta$MOT reconstruction based on HSE = 0.5. **(c)** Our reconstructed $\Delta$MOT based on HSE = 0.1 (falls under the corresponding regression line). The decrease in uncertainties on our $\Delta$MOT reconstructions reflects the decrease in uncertainty on our scaled $\Delta$GMSST reconstructions. **(d–f)** Our reconstructed global mean ocean temperature anomaly ($\Delta$MOT) for 1.5–4.5 Ma based on three different HSEs (orange lines) compared to the 2 Myr smoothed $\Delta$BWT reconstruction from Cramer et al. (2011) using their Eq. (7b) (red line) with 90 % confidence interval (orange shading). Their BWTs were mean-shifted by 2.1 °C so that they are the same as the long-term (401 kyr running average) mean as our $\Delta$MOT reconstruction over the last 400 kyr. **(d)** Our reconstructed $\Delta$MOT based on HSE = 1. **(e)** Our reconstructed $\Delta$MOT based on HSE = 0.5. Also shown are $\Delta$MOT reconstructions based on HSE = 0.7 and HSE = 0.3 (brown lines), which closely encompass the $1\sigma$ uncertainty on our $\Delta$MOT reconstruction based on HSE = 0.5. **(f)** Our reconstructed $\Delta$MOT based on HSE = 0.1 (falls under the corresponding regression line).

erage) (Clark et al., 2024), as well as an estimate from a global climate model with data assimilation of −3.1 °C (−3.4 to −2.9 °C, 95 % CI) (23–19 ka average) (Tierney et al., 2020), with these reconstructions suggesting an average LGM $\Delta$GMSST of −3.0 ± 0.2 °C.

There is also good agreement among existing $\Delta$MOT and $\Delta$DOT reconstructions for the LGM (Fig. 7l). The $\Delta$DOT reconstructions by Hansen et al. (2013, 2023) are based on an inferred LGM cooling of −2° and are thus not considered here. Shakun et al. (2015) found an LGM average cool-

ing of −2.63 °C, and Rohling et al. (2022) (updated from Rohling et al., 2021) reconstructed an LGM average cooling of −2.82 ± 0.17 °C. Shackleton et al. (2023) standardized previously published ice-core noble-gas $\Delta$MOT reconstructions (Bereiter et al., 2018; Baggenstos et al., 2019; Shackleton et al., 2019, 2020) and reported a data-based LGM average $\Delta$MOT of −2.76 ± 0.27 °C and a spline average of −2.83 ± 0.5 °C. We also note that a $\Delta$DOT reconstruction for 0–20 ka derived by subtracting global $\delta^{18}O_{sw}$ from $\delta^{18}O_b$ records (Zhu et al., 2024) is in good agreement with

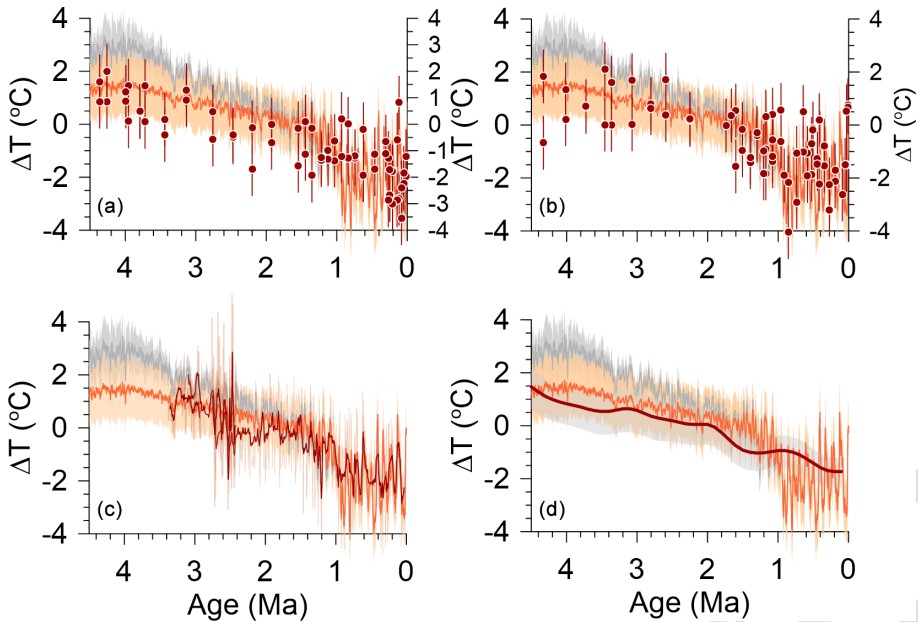

**Figure 6.** Our reconstructed change in mean ocean temperature (ΔMOT) based on HSE = 0.5 before 1.5 Ma (1σ uncertainty) and HSE = 1 after 0.9 Ma with a linear increase in HSE from 0.5 to 1 between 1.5 and 0.9 Ma (orange) compared to reconstructions of bottom water temperature anomalies (ΔBWT) from Mg / Ca proxies. Also shown is ΔMOT based on HSE = 1 (gray) for the past 4.5 Ma. **(a)** ΔBWT from equatorial Pacific ODP site 806 (Lear et al., 2003). Mg / Ca BWTs were mean-shifted by −1.73 °C to fall within same range as our ΔMOT reconstruction for the last 800 kyr. **(b)** ΔBWT from equatorial Atlantic ODP site 926 (Lear et al., 2003). Mg / Ca BWTs were mean-shifted by −1.73 °C so that they are referenced to the Holocene (i.e., 0 °C for the last 10 kyr). **(c)** ΔBWT from North Atlantic DSDP site 607 for > 2.9 Ma (Dwyer and Chandler, 2009) and < 2.8 Ma (Sosdian and Rosenthal, 2009) (11-point running average in dark red) and ΔBWT from North Atlantic IODP site U1313 for 2.4–2.75 Ma (Jakob et al., 2020). Site 607 data are mean-shifted by 2.71 °C so that they are referenced to the Early Holocene (i.e., 0 °C at 10 ka). Site U1313 data are referenced to modern BWT derived by Mg / Ca measurements on core-top sample. **(d)** Smoothed ΔBWT reconstruction from Cramer et al. (2011) (dark-red line) using their Eq. (7b) with 90 % confidence interval (gray shading). BWTs were mean-shifted by 2.1 °C so that they have the same long-term mean as our ΔMOT reconstruction over the last 400 kyr. The 1σ uncertainty on our ΔMOT reconstructions is derived by combining in quadrature 1σ uncertainties on the ΔGMSST reconstruction and on the relationship between ΔGMSST and ΔMOT (Fig. 2a), which closely corresponds to a range in HSE between 0.7 and 0.3 (Fig. 6b, e).

the spline ΔMOT reconstruction (Shackleton et al., 2023), including during the period of LGM overlap (18–20 ka) (Fig. 7l). Taking the average LGM ΔMOT (−2.76 ± 0.31 °C) with the average LGM ΔGMSST (−3.0 ± 0.2 °C) suggests an LGM HSE of 0.9, or the same as when using the ice-core noble-gas ΔMOT reconstructions for the last 0.7 Myr (Fig. 7c) and, within uncertainty, with an HSE of ∼ 1 suggested by climate models (Fig. 2).

Recent work, however, has argued that changes in noble-gas saturation states of the deep ocean during the LGM may have resulted in a cold bias in the ice-core data equivalent to −0.38 ± 0.37 °C (Seltzer et al., 2024) to −0.50 ± 0.67 °C (Pöppelmeier et al., 2023). Despite these similar results, Seltzer et al. (2024) noted that Pöppelmeier et al. (2023) were unable to reconcile their result from simulated air–sea gas exchange with the ice-core data and thus based their MOT cooling on a climate model. The cold bias identified by Seltzer et al. (2024) is based on an ensemble of five PMIP3 climate models that suggest air–sea disequilibria due to stronger high-latitude winds, from which they argued

that LGM ΔMOT was −2.27 ± 0.46 °C, corresponding to an LGM HSE of ∼ 0.8 when using an average LGM ΔGMSST of −3.0 ± 0.2 °C. Seltzer et al. (2024) did not elaborate on how this effect may have differed at other times, but we can assume that high-latitude winds would have weakened from their LGM maxima, resulting in a diminished effect on air–sea disequilibria at those times. As a first-order approximation of how this would be expressed during other glacial maxima, we decreased all glacial maxima ΔMOTs in the 0.7 Myr ice-core noble-gas reconstruction (Haeberli et al., 2021) by 0.38 °C, resulting in a decrease in HSE from 0.9 to 0.85.

We thus conclude that the potential LGM cold bias in ΔMOT does not substantially change the evidence that HSE was ∼ 1 within the uncertainties of the data. At the same time, Seltzer et al. (2024) recognized that current understanding of changes in LGM high-latitude wind speed (and thus air–sea disequilibria) in both observations and models is highly uncertain. For example, more-recent modeling suggests that high-latitude Southern Hemisphere wind speeds during the LGM weakened by 15 % (Zhu et al., 2021) to

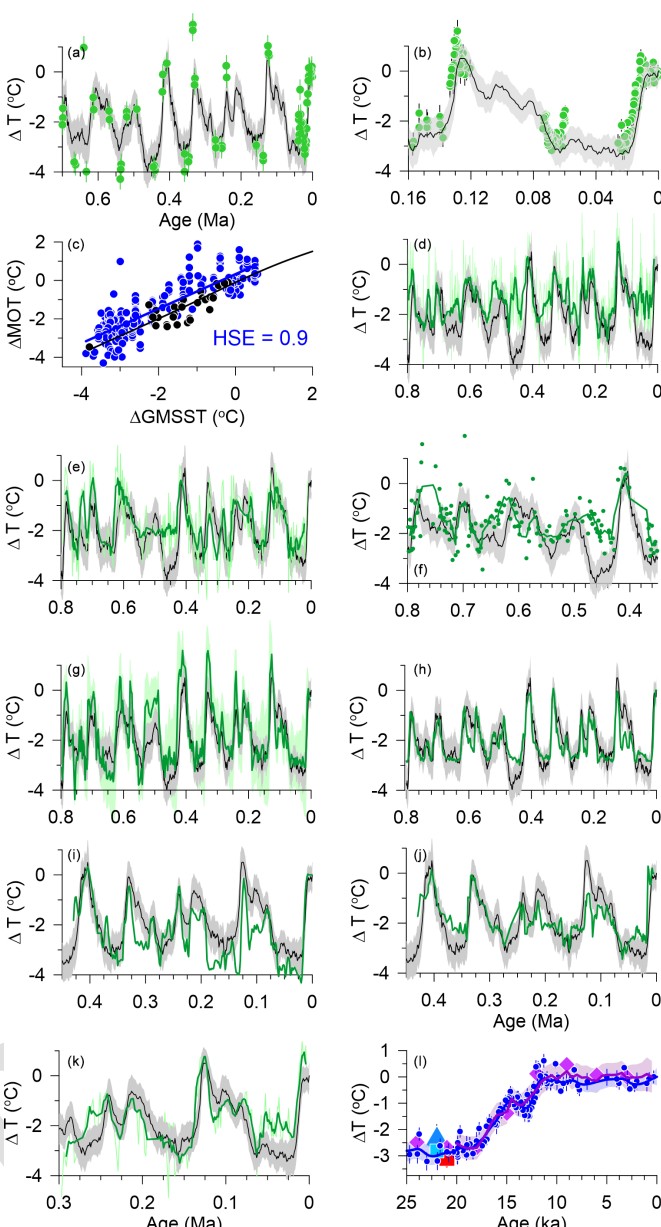

**Figure 7.** Comparison of the reconstruction of global mean sea surface temperature change (Clark et al., 2024) (black) to proxy reconstructions of bottom water and mean ocean temperature change from preindustrial (green). **(a)** ΔMOT reconstruction from Haeberli et al. (2021). **(b)** ΔMOT reconstructions from Shackleton et al. (2020, 2021, 2023). **(c)** Blue: scatter plot of ΔMOT derived from ice-core noble gases (Bereiter et al., 2018; Haeberli et al., 2021; Shackleton et al., 2019, 2020, 2021) versus the reconstruction of global mean sea surface temperature change from preindustrial for the last 0.7 Myr (Clark et al., 2024). Linear regression through the data identifies an HSE of 0.9. Also shown are climate model data (black circles) for ΔGMSST < 0 °C with smoothing spline fit (black line) (see Fig. 2). **(d)** Mg / Ca-based ΔBWT reconstruction from site 1123 for the last 0.8 Ma (Elderfield et al., 2012) (11-point running average shown by dark-green line). **(e)** Mg / Ca-based ΔBWT reconstruction modified from Sosdian and Rosenthal (2009) (5-point running average shown by dark-green line). **(f)** Mg / Ca-based ΔBWT reconstruction from ODP site 1208 from Ford and Raymo (2020) (5-point running average shown by green line). **(g)** ΔBWT reconstruction from Shakun et al. (2015). **(h)** ΔBWT reconstruction from Rohling et al. (2022). **(i)** ΔBWT reconstruction from ODP site 980 from Waelbroeck et al. (2002). **(j)** ΔBWT reconstruction from site 94-10 from Waelbroeck et al. (2002). **(k)** Mg / Ca-based ΔBWT reconstruction from Martin et al. (2002) (5-point running average shown by dark-green line). **(l)** Reconstructed changes in ΔGMSST, ΔMOT, and ΔDOT since the Last Glacial Maximum. Blue symbols with 1σ uncertainty are ΔMOT derived from ice-core noble gases, with spline fit to these data with 1σ uncertainty shown by the blue curve and 1σ confidence envelope (Shackleton et al., 2023). Purple diamonds are ΔDOT from Shakun et al. (2015). Light-blue square is ΔDOT from Rohling et al. (2022). Thick purple line with 1σ confidence envelope is ΔDOT from Zhu et al. (2024). Light-blue triangle with 1σ uncertainty is ΔMOT from Seltzer et al. (2024). Red square with 1σ uncertainty is ΔGMSST from Tierney et al. (2020).

25 % (Gray et al., 2023) as opposed to the $11 \pm 23\%$ increase in the PMIP3 model average (largely driven by one model). As Seltzer et al. (2024) note, "Future improvements in our understanding of high-latitude winds in the LGM will help to substantially reduce uncertainties in LGM MOT". In the meantime, the good agreement between independent estimates of $\Delta$DOT from marine proxy records (Shakun et al., 2015; Rohling et al., 2022; Zhu et al., 2024) with the ice-core $\Delta$MOT estimate that neglects air–sea disequilibrium (Shackleton et al., 2023) (Fig. 2c) suggests a negligible effect of air–sea disequilibrium on the ice-core data.

## 2.3 Summary of changes in MOT and HSE over the past 4.5 million years

In summary, existing constraints from Mg / Ca-based $\Delta$BWT data and ice-core $\Delta$MOT data show that HSE was $\sim 0.5 \pm 0.2$ for $> 1.5$ Ma when $\Delta$GMSSTs were $> 0\,°C$ and $\sim 1$ for the past 0.8 Myr when $\Delta$GMSSTs were $< 0\,°C$ (Fig. 8), consistent with climate model results (Fig. 2). These constraints thus suggest that the increase occurred as part of the large-scale changes in ocean circulation during the MPT (0.9–1.5 Ma) (Lisiecki, 2014; Lang et al., 2016; Pena and Goldstein, 2014; Hodell and Venz-Curtis, 2006). Although the data do not identify the exact function by which HSE increased during this time, we make the simplifying assumption that it increased linearly from 0.5 to 1 during the MPT (Fig. 8). As previously noted, HSE increases of 0.3 to 1 and 0.7 to 1 result in $\Delta$MOTs that fall within the $1\sigma$ uncertainty of $\Delta$MOTs based on an increase of 0.5 to 1 (Figs. 5, 8, S2). We also find that our $\Delta$MOT reconstruction is insensitive to a shorter (0.9–1.2 Ma) or longer (0.7–1.7 Ma) interval over which HSE increased from 0.5 to 1 (Fig. S2C).

## 3 Decomposition of the benthic $\delta^{18}$O record

### 3.1 Derivation of $\delta^{18}$O$_{sw}$ supports HSE < 1 before MPT

Using our $\Delta$MOT reconstruction to decompose the $\delta^{18}$O$_b$ record into its temperature ($\delta^{18}$O$_T$) and seawater ($\delta^{18}$O$_{sw}$) components provides additional support for the need to decrease HSE to $< 1$ prior to 0.9 Ma as identified from the proxy $\Delta$BWT data. We use the probabilistic global $\delta^{18}$O$_b$ stack (Prob-stack) from Ahn et al. (2017), which, compared to the previous LR04 stack (Lisiecki and Raymo, 2005), includes a larger number of records (total $= 180$) that span a larger depth range of the ocean (500–4500 m) as well as uncertainties in the alignment of these records. Stacking individual $\delta^{18}$O$_b$ records that span such a geographic and depth range significantly increases the signal-to-noise ratio for global changes in $\delta^{18}$O$_b$ and its $\delta^{18}$O$_T$ and $\delta^{18}$O$_{sw}$ components while minimizing local hydrographic changes that may contribute to a single $\delta^{18}$O$_b$ record. We convert our $\Delta$MOT reconstruction (Fig. 8) to $\delta^{18}$O$_T$ using a relation of $0.25\,‰\,°C^{-1}$, which is appropriate for cold deepwater tem-

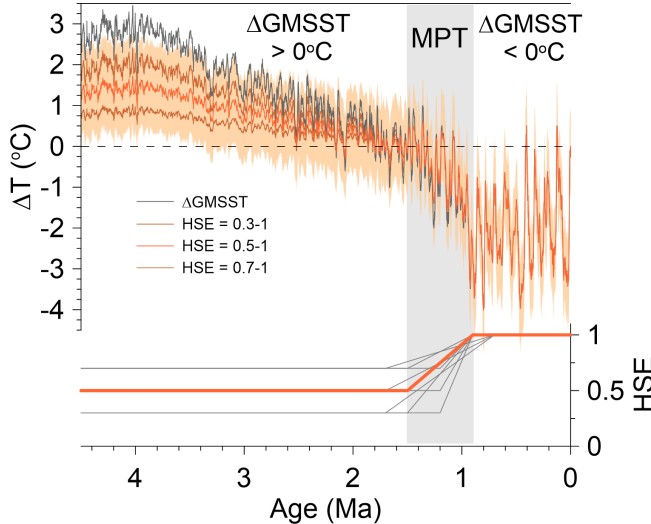

**Figure 8.** Our reconstructed $\Delta$MOT (orange line with $1\,\sigma$ uncertainty) that linearly increases from 0.5 to 1 across the MPT. Also shown is $\Delta$GMSST (black line) from (Clark et al., 2024) and $\Delta$MOTs based on increases from 0.3 to 1 and 0.7 to 1 across the MPT (brown lines). Lower panel shows different scenarios (magnitude and duration) for the increase in HSE assessed in Fig. S2, with our preferred scenario shown by the orange line.

peratures (Kim and O'Neil, 1997; Marchitto et al., 2014), and subtract this from the Prob-stack to derive $\delta^{18}$O$_{sw}$. We estimate the $1\sigma$ uncertainty on $\delta^{18}$O$_{sw}$ from the square root of sum of squares based on the uncertainties in our $\Delta$MOT reconstruction, the HSE from models, and the probabilistic $\delta^{18}$O$_b$ stack.

Figure 9a shows that when using an HSE of 1 for the last 4.5 Myr, Early Pleistocene and Late Pliocene $\delta^{18}$O$_{sw}$ values are significantly more positive than the Mg / Ca-based reconstruction from North Atlantic site 607 (Sosdian and Rosenthal, 2009), which is the longest (0–3.2 Ma), orbitally resolved record available. This is to be expected since the Mg / Ca-based $\Delta$BWT reconstruction was one of several used to constrain the reduction in HSE. However, positive $\delta^{18}$O$_{sw}$ values extend back to 4.5 Ma, with Pliocene interglacial values that are 0.3 ‰ to 0.5 ‰ more positive than average interglacial values over the last 0.8 Myr despite robust evidence for higher-than-present Pliocene sea levels that require values below 0 ‰ (Miller et al., 2012; Raymo et al., 2018; Dumitru et al., 2019; Winnick and Caves, 2015). This suggests that too much of the $\delta^{18}$O$_b$ signal is being removed by the $\delta^{18}$O$_T$ component using an HSE of 1.

Applying our $\Delta$MOT reconstruction (Fig. 8) results in $\delta^{18}$O$_{sw}$ values that are more comparable to Pliocene sea-level reconstructions, with average Pliocene interglacial $\delta^{18}$O$_{sw}$ values decreasing to 0 ‰ to 0.1 ‰ (Fig. 9b), thus supporting the need for the decrease in HSE prior to 1.5 Ma suggested by proxy data. However, Early Pleistocene and Pliocene $\delta^{18}$O$_{sw}$ values continue to be more positive than data con-

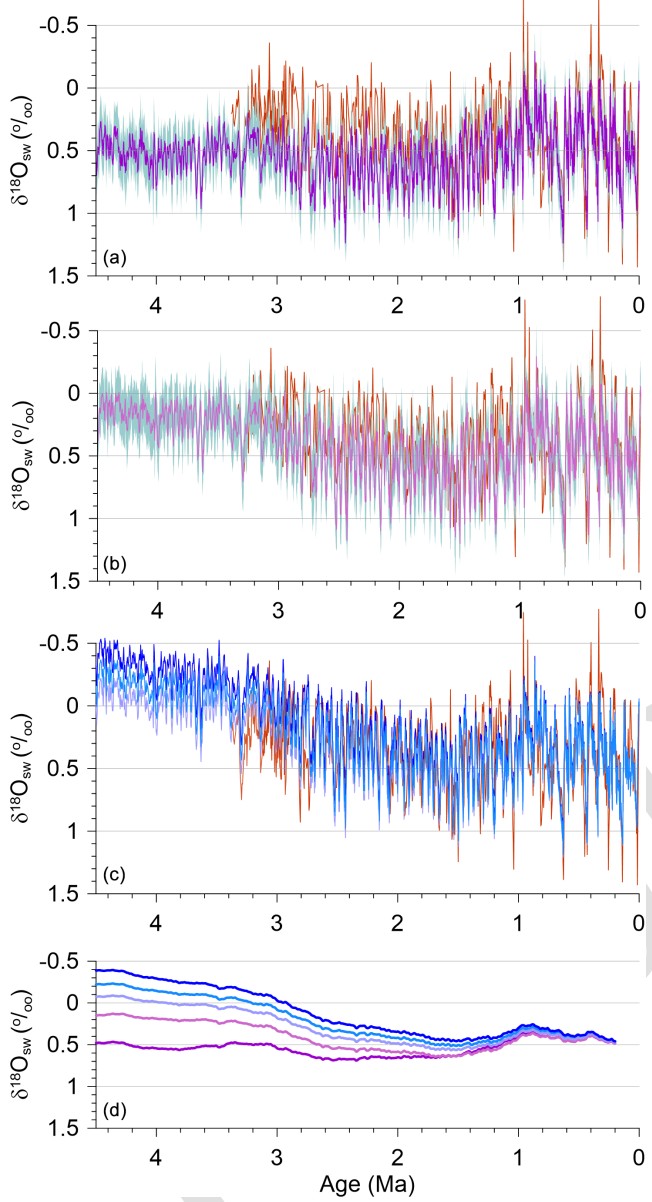

**Figure 9. (a)** Our reconstructed $\delta^{18}O$ of seawater ($\delta^{18}O_{sw}$) ($1\sigma$ uncertainty) for 0–4.5 Ma based on HSE $= 1$ (purple) compared to proxy-based (Mg / Ca) reconstruction of $\delta^{18}O_{sw}$ (red) from North Atlantic site 607 (Sosdian and Rosenthal, 2009; Dwyer and Chandler, 2009). **(b)** Our reconstructed $\delta^{18}O$ of seawater ($\delta^{18}O_{sw}$) ($1\sigma$ uncertainty) for 0–4.5 Ma based on HSE $= 0.5$ before 1.5 Ma (violet) compared to proxy-based (Mg / Ca) reconstruction of $\delta^{18}O_{sw}$ (red) from North Atlantic site 607 (Sosdian and Rosenthal, 2009; Dwyer and Chandler, 2009). **(c)** Our reconstructed $\delta^{18}O$ of seawater ($\delta^{18}O_{sw}$) ($1\sigma$ uncertainty) for 0–4.5 Ma based on HSE $= 0.5$ before 1.5 Ma and removal of long-term trends of 0.05 ‰ Myr$^{-1}$ ($\delta^{18}O_{sw}$-LO, light blue), 0.083 ‰ Myr$^{-1}$ ($\delta^{18}O_{sw}$-INT, medium blue), and 0.12 ‰ Myr$^{-1}$ ($\delta^{18}O_{sw}$-HI, dark blue) compared to proxy-based (Mg / Ca) reconstruction of $\delta^{18}O_{sw}$ (red) from North Atlantic site 607 (Sosdian and Rosenthal, 2009; Dwyer and Chandler, 2009). **(d)** Long-term running averages of the five $\delta^{18}O_{sw}$ scenarios shown in panels **(a)**–**(c)**, color coded in the same way.

straints (Dwyer and Chandler, 2009; Dumitru et al., 2019; Miller et al., 2012; Jakob et al., 2020; Sosdian and Rosenthal, 2009). While further decreases in HSE result in more negative Pliocene $\delta^{18}O_{sw}$ values (Fig. S2B), it would have to decrease to $\sim 0.1$ to achieve average Pliocene $\delta^{18}O_{sw}$ values of $-0.2$ ‰ (not shown). Although this could explain higher sea levels at that time (Raymo et al., 2018; Winnick and Caves, 2015; Dumitru et al., 2019), such a low HSE is ruled out by proxy BWT reconstructions (Figs. 4, 5, 6).

## 3.2 Other potential factors influencing the long-term $\delta^{18}O_b$ trend

Recent studies have identified offsets between Cenozoic ocean temperatures inferred from the $\delta^{18}O_b$ record as compared to proxy-based temperatures, notably those based on clumped isotope thermometry, and considered possible effects on $\delta^{18}O_b$ such as from diagenesis, carbonate ion changes, or changes in ocean pH; issues with the calibration of the proxy itself may also be involved (Meckler et al., 2022; Rohling et al., 2024). Raymo et al. (2018) suggested that the $\sim 0.3$ ‰ decrease in Pliocene $\delta^{18}O_b$ values relative to Late Holocene values is too small to accommodate both the higher sea levels and warmer ocean temperatures inferred for this time. They addressed this discrepancy by proposing that foraminifera tests that recrystallized in pore waters that were colder than those in which they were originally buried would cause precipitation of abiotic calcite with heavier $\delta^{18}O$ values (Schrag, 1999), resulting in 3 Myr benthic foraminifera tests being $\sim 0.25$ ‰ heavier than tests with no diagenesis. Decreasing Pliocene $\delta^{18}O_b$ values in the Prob-stack by an additional 0.25 ‰ can then more readily explain the evidence for higher sea levels and warmer ocean temperatures. Applying this diagenetic process using our reconstructed MOT cooling over the past 4.5 Myr suggests that the diagenetic impact is subtle but could account for a shift of 0.2 ‰ to 0.4 ‰ over the 4.5 Myr record, with averaging of the $\delta^{18}O_b$ records in the Prob-stack integrating diagenetic effects to produce a near-constant long-term trend.

We cannot determine exactly how much the effect of diagenesis may have contributed to an increase in $\delta^{18}O_b$ over the least 4.5 Myr, but our assessment suggests that it is sufficient to justify a long-term increase as proposed by Raymo et al. (2018). Given our assessed range, we considered three scenarios that result in a diagenetic increase in $\delta^{18}O_b$ from today of 0.15 ‰, 0.25 ‰, and 0.35 ‰ at 3 Ma, corresponding to a long-term increase of 0.05 ‰ Myr$^{-1}$ ($\delta^{18}O_{sw}$-LO), 0.083 ‰ Myr$^{-1}$ ($\delta^{18}O_{sw}$-INT), and 0.12 ‰ Myr$^{-1}$ ($\delta^{18}O_{sw}$-HI), respectively. In removing the long-term increase for each of these scenarios from the $\delta^{18}O_b$ Prob-stack, $\delta^{18}O_{sw}$ values based on our $\Delta$MOT reconstruction agree with reconstructed Late Pliocene and Early Pleistocene $\delta^{18}O_{sw}$ values from site 607 (Fig. 9c). Although this agreement hinges on just one $\delta^{18}O_{sw}$ record that may also have experienced diagenesis, we previously showed that this site closely monitors

MOT (Fig. 4). Moreover, not accounting for these combined effects (change in HSE, diagenesis) would result in long-term average $\delta^{18}O_{sw}$ values being comparable to Late Pleistocene average values throughout the last 4.5 Myr (Fig. 9d). Finally, our reconstructed long-term Pliocene $\delta^{18}O_{sw}$ values of $-0.1\,‰$ to $-0.4\,‰$ (Fig. 9d) are consistent with sea-level highstands of 20–25 m above present (Dumitru et al., 2019) even when accounting for land ice having more positive $\delta^{18}O$ values under warmer surface temperatures (Winnick and Caves, 2015), with the remaining $\sim 0.4\,‰$ decrease in the $\delta^{18}O_b$ record relative to the Late Pleistocene consistent with the $\sim 1.5\,°C$ $\Delta$MOT in our reconstruction (Fig. 8). We also evaluated a scenario of having the same magnitude of diagenetic correction (0.375 ‰) over the past 4.5 Myr but with the increase tracking our $\Delta$MOT reconstruction. Figure S3 shows that the faster rate of change across the MPT slightly increases the trend in $\delta^{18}O_{sw}$ over this interval (more depleted before 1 Ma, more enriched after 1 Ma), but the changes are insignificant ($\leq 0.1\,‰$).

Another potential effect that may be comparable to diagenesis is the impact of changing carbonate ion concentration in seawater on the $\delta^{18}O$ of foraminifera shells (Spero et al., 1997). Due to higher concentrations of atmospheric $CO_2$ (Köhler, 2023), the 100 kyr mean carbonate ion concentration in Pliocene surface seawater may have been lower than during the Late Pleistocene by on the order of 20 to 50 $\mu$mol kg$^{-1}$. Laboratory experiments (Bijma et al., 1999) and theoretical studies (Zeebe, 1999) on planktic species suggest a species-specific effect which would cause the $\delta^{18}O$ of foraminifera shells in the Early Pliocene to increase by 0.1 ‰ to 0.3 ‰ relative to the Late Pleistocene. Although no laboratory studies on benthic foraminifera species have been conducted, Marchitto et al. (2014) speculated that the $\delta^{18}O$ of some Late Holocene benthic species may have been influenced by pH. On the other hand, a recent study compiling 160 kyr of data from two widely abundant planktic foraminifera species in wider tropical surface waters extracted from 127 sediment cores could not confirm the carbonate ion effect as found in the laboratory (Köhler and Mulitza, 2024). Furthermore, a recent analysis of Late Holocene data from the benthic foraminifera species *Cibicidoides* spp. found that only about a third of the variance in $\delta^{18}O$ can be explained by carbonate chemistry (Nederbragt, 2023). Current understanding of the carbonate ion effect thus remains uncertain but suggests that it may have contributed to the $\delta^{18}O$ corrections in addition to diagenesis. Regardless of the specific combination of potential effects, our results clearly identify the need to account for a long-term increase in $\delta^{18}O_b$ when deriving $\delta^{18}O_{sw}$ and aligning ocean temperature records (Rohling et al., 2024; Meckler et al., 2022).

## 3.3 Temperature and $\delta^{18}O_{sw}$ controls on the Prob-stack $\delta^{18}O_b$ record

Our reconstruction shows that $\delta^{18}O_{sw}$ increased between 3.0 and 2.5 Ma in its (glacial) maxima to values that, on average, are similar to LGM values. These high glacial values in $\delta^{18}O_{sw}$ persisted throughout much of the Pleistocene (Fig. 10b). The main factor that is modulating the expression of this Late Pliocene/Early Pleistocene $\delta^{18}O_{sw}$ increase in the Prob-stack $\delta^{18}O_b$ is the gradual decrease in long-term average MOT (and thus increase in $\delta^{18}O_T$) relative to the increase in the rate of change in long-term average $\delta^{18}O_b$ values between 3.0–2.5 Ma (Fig. 10b). In other words, since $\delta^{18}O_T$ is only decreasing gradually, the relatively rapid increase in $\delta^{18}O_b$ between 3.0–2.5 Ma must be due to a substantial increase in $\delta^{18}O_{sw}$. After 2.5 Ma, both average $\delta^{18}O_T$ and $\delta^{18}O_{sw}$ values increase at similar rates until the onset of the MPT at 1.5 Ma. The second rapid increase in $\delta^{18}O_b$ during the MPT would then be due to the rapid increase in $\delta^{18}O_T$, in this case resulting in a slight decrease in average $\delta^{18}O_{sw}$ values (Fig. 10b). A subsequent paper will use our $\delta^{18}O_{sw}$-LO, $\delta^{18}O_{sw}$-INT, and $\delta^{18}O_{sw}$-HI reconstructions to derive sea level over the last 4.5 Myr that accounts for land ice having more positive $\delta^{18}O$ values under warmer surface temperatures (Winnick and Caves, 2015; Gasson et al., 2016).

## 3.4 Assessment of our $\delta^{18}O_{sw}$-INT reconstruction

We assess our $\delta^{18}O_{sw}$-INT reconstruction by comparing them with other $\delta^{18}O_{sw}$ reconstructions that have been derived from two methods. One method is directly comparable to ours in having used an independent reconstruction of BWT which is then subtracted as $\delta^{18}O_T$ from the $\delta^{18}O_b$ record to derive $\delta^{18}O_{sw}$ (Shakun et al., 2015; Sosdian and Rosenthal, 2009; Elderfield et al., 2012; Woodard et al., 2014; Ford and Raymo, 2020; Miller et al., 2020). Other than the Shakun et al. (2015) and Miller et al. (2020) reconstructions, these are based on local records and are thus subject to local temperature and hydrographic effects, and the small number of high-resolution records prevents development of a robust global stack. As summarized in Sect. 2.2.1, the other method takes the opposite approach of ours by first reconstructing sea level and then subtracting it as $\delta^{18}O_{sw}$ from the $\delta^{18}O_b$ record to derive $\Delta$BWT or $\Delta$DOT.

Figure 11 compares our $\delta^{18}O_{sw}$-INT reconstruction with published reconstructions for the last 0.8 Myr, with Fig. 11a–e comparing reconstructions derived from existing BWT reconstructions (Ford and Raymo, 2020; Miller et al., 2020; Shakun et al., 2015; Sosdian and Rosenthal, 2009; Elderfield et al., 2012) and Fig. 11f comparing one derived from a regression-based sea-level reconstruction (Rohling et al., 2022). In general, there is good agreement between the BWT-derived reconstructions and our reconstruction, although there are some differences with reconstructions based

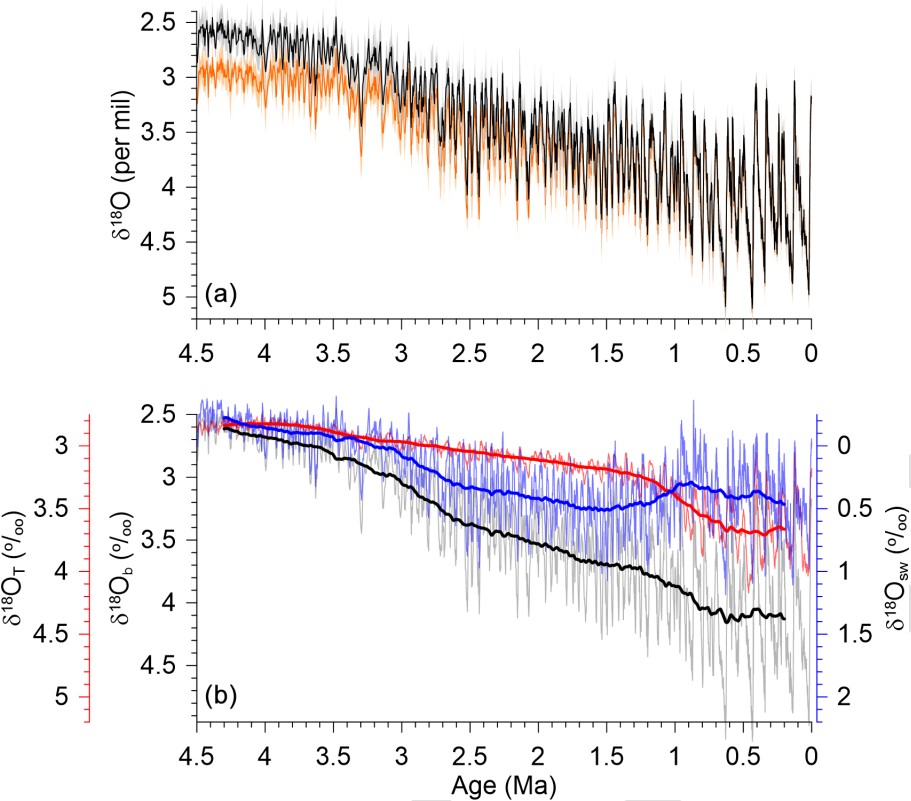

**Figure 10. (a)** Comparison of the Prob-stack $\delta^{18}O_b$ record (orange line with $1\sigma$ uncertainty) (Ahn et al., 2017) to the Prob-stack $\delta^{18}O_b$ record with the removal of a long-term increase of $0.083\%_o\,\mathrm{Myr^{-1}}$ (black line with $1\sigma$ uncertainty). **(b)** Comparison of the Prob-stack $\delta^{18}O_b$ record (Ahn et al., 2017) with the removal of a long-term increase of $0.083\%_o\,\mathrm{Myr^{-1}}$ (gray with 251-point running average in black) to our $\delta^{18}O_T$ reconstruction (light red with 251-point running average in red) and our $\delta^{18}O_{sw}$-INT reconstruction (light blue with 251-point running average in blue).

on individual $\delta^{18}O_b$ records during interglaciations, particularly at site 1208 (Fig. 11e) (Ford and Raymo, 2020), which we attribute to some combination of differences in site-specific $\delta^{18}O_T$ and $\delta^{18}O_{sw}$. If the sea-level-based $\delta^{18}O_{sw}$ reconstruction (Rohling et al., 2021; Rohling et al., 2022) is a global signal not affected by local temperature or hydrography, then the high agreement between the two reconstructions (Fig. 11f) that were derived by completely independent means provides very high confidence in them. We also note that our LGM (19–26 ka) change in $\delta^{18}O_{sw}$ from modern values ($0.9 \pm 0.1\%_o$) is in agreement with a pore-water-based reconstruction ($1.0 \pm 0.1\%_o$) (Schrag et al., 1996, 2002).

Figure 12 compares our $\delta^{18}O_{sw}$-INT reconstruction with published reconstructions that span some part or all of the Late Pliocene and Early Pleistocene. There is a notable difference with the sea-level-based reconstruction for times older than 0.9 Ma (Rohling et al., 2021; Rohling et al., 2022), which is primarily expressed by glacial $\delta^{18}O_{sw}$ values being substantially less positive than in our reconstruction, particularly between 3–0.9 Ma (Fig. 12a). We attribute this difference to the previously noted problem with the regression approach used by Rohling et al. (2022, 2021) to reconstruct

sea level that preserves the variability of the $\delta^{18}O_b$ record, including the increase in amplitude during the MPT. The Miller et al. (2020) $\delta^{18}O_{sw}$ reconstruction (Fig. 12b) used the smoothed BWT record from Cramer et al. (2011) to extract the $\delta^{18}O_{sw}$ signal, thus assuming that most of the orbital-scale $\delta^{18}O_b$ variability is comprised of $\delta^{18}O_{sw}$. This results in their $\delta^{18}O_{sw}$ reconstruction being in reasonable agreement with our $\delta^{18}O_{sw}$-INT reconstruction, although their Early Pleistocene $\delta^{18}O_{sw}$ glacial values tend to be $0.25\%_o$–$0.5\%_o$ more negative than in our reconstruction (Fig. 12b). Figure 12c again compares BWT-based $\delta^{18}O_{sw}$ reconstructions from site 607 (Dwyer and Chandler, 2009; Sosdian and Rosenthal, 2009) that we previously used to gauge the sensitivity of our $\delta^{18}O_{sw}$ reconstruction to changes in HSE and long-term trends in $\delta^{18}O_b$ (Fig. 9). The ostracode-based values for site $607 > 3$ Ma are on average slightly more positive than our reconstruction (Fig. 12c), which may reflect uncertainties in the Mg / Ca temperature calibration that requires validation. Otherwise, we emphasize the good agreement between records in their orbital-scale variability. Finally, Fig. 12d compares our reconstruction to two reconstructions from North Atlantic site U1313 (Jakob et al., 2020) that differ

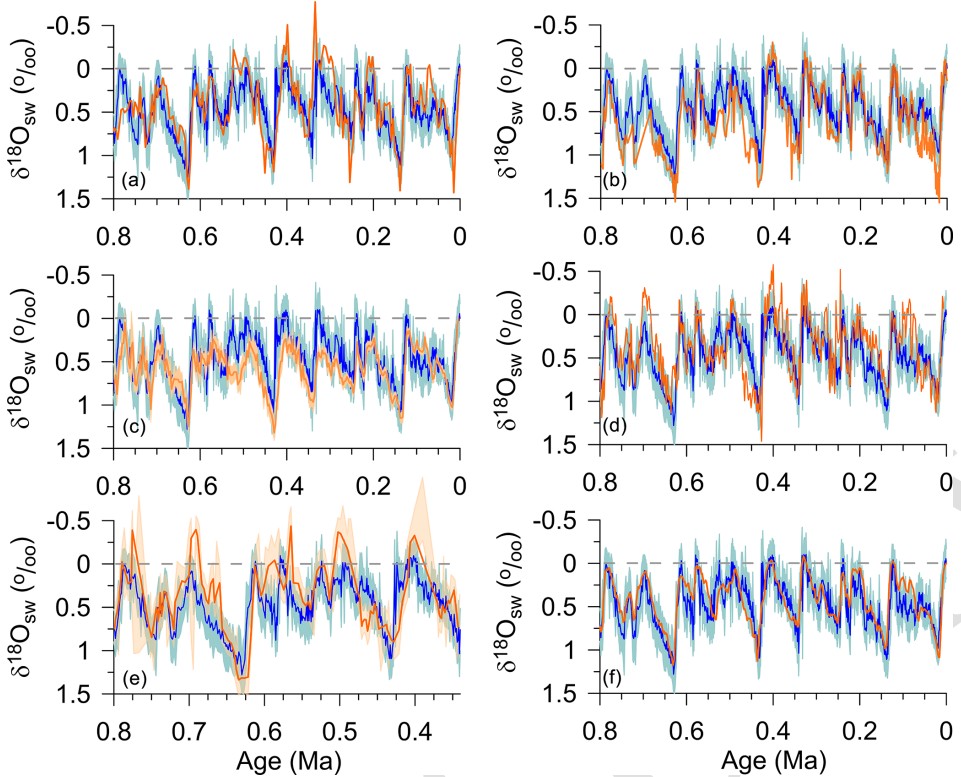

**Figure 11.** Comparison of our $\delta^{18}O_{sw}$-INT reconstruction (blue) to published $\delta^{18}O_{sw}$ reconstructions (orange). **(a)** Mg / Ca-based $\delta^{18}O_{sw}$ reconstruction from Sosdian and Rosenthal (2009). **(b)** $\delta^{18}O_{sw}$ reconstruction from Miller et al. (2020). **(c)** Proxy-based $\delta^{18}O_{sw}$ reconstruction from Shakun et al. (2015). **(d)** Mg / Ca-based $\delta^{18}O_{sw}$ reconstruction from Elderfield et al. (2012). **(e)** Mg / Ca-based $\delta^{18}O_{sw}$ reconstruction from Ford and Raymo (2020). **(f)** $\delta^{18}O_{sw}$ reconstruction from Rohling et al. (2022). We note that our LGM (19–26 ka) change in $\delta^{18}O_{sw}$ from modern CE5 is $0.9 \pm 0.1$ ‰, in agreement with a pore-water-based reconstruction of $1.0 \pm 0.1$ ‰ (Schrag et al., 1996, 2002).

based on their Mg / Ca calibrations. As with nearby site 607, this record shows good agreement with our reconstruction in the orbital-scale variability, with differences in amplitude reflecting the different Mg / Ca calibrations.

We next compare our $\Delta$MOT and $\delta^{18}O_{sw}$ reconstructions to those from Pacific sites 1123 and 1208 for two intervals when there are substantial differences between their reconstructions and ours (Fig. 13). Of the nine $\Delta$MOT and $\Delta$BWT reconstructions we had previously compared our $\Delta$MOT reconstruction to for some or all of the last 0.7 Myr (Fig. 7), these two sites showed the largest differences, with site 1123 having good agreement in its temporal variability but having substantially warmer glacial intervals (Fig. 7d), while site 1208 showed some differences in the timing and amplitude of its variability (Fig. 7f). Figure 13a and c also show that temperatures at sites 1123 and 1208 are significantly different than our $\Delta$MOT reconstruction between 0.9–1.4 Ma which spans much of the MPT.

These times of temperature differences between the two Pacific sites and our $\Delta$MOT reconstruction result in significant differences in their site-specific $\delta^{18}O_{sw}$ values and our global $\delta^{18}O_{sw}$ reconstruction. Elderfield et al. (2012) and

Ford and Raymo (2020) argued that the more positive $\delta^{18}O_{sw}$ values at sites 1123 and 1208 after 0.9 Ma suggest an increase in ice-sheet volume. Two factors, however, suggest that these changes may instead reflect regional hydrographic changes. The first is that $\Delta$BWTs and $\delta^{18}O_{sw}$ values at the start of the site 1123 record (1.55 Ma) are similar to our reconstructions until ~ 1.4 Ma, when they depart from our reconstructions until ~ 0.9 Ma, when they then merge again with our reconstructions (Fig. 13a, b). This implies that site 1123 is recording large ice sheets at ~ 1.5 Ma, or before the MPT. The second aspect that suggests that the more-negative $\delta^{18}O_{sw}$ values at these two sites between 1.4–0.9 Ma are not representative of global values is based on their implications for sea-level change. As a first-order approximation, we scale $\delta^{18}O_{sw}$ to sea level using $0.008$ ‰ m$^{-1}$ as derived from LGM pore-water (Schrag et al., 2002) and sea-level (Lambeck et al., 2014) estimates CE6. The scaled pre-MPT $\delta^{18}O_{sw}$ values at the two sites would lead to sea-level highstands that are 50–100 m higher than present throughout much of the 0.9–1.4 Ma interval (Fig. 13b, d), thus implying an essentially ice-free world. We thus conclude that the differences in $\Delta$BWT and $\delta^{18}O_{sw}$ at sites 1123 and 1208 from global values be-

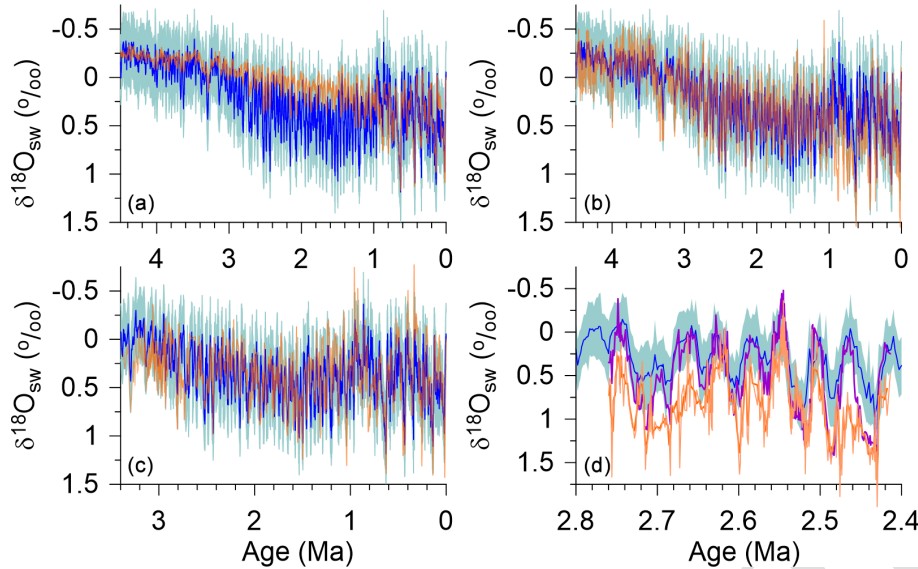

**Figure 12.** Comparison of our $\delta^{18}O_{sw}$-INT reconstruction (blue) to published $\delta^{18}O_{sw}$ reconstructions (orange). **(a)** $\delta^{18}O_{sw}$ reconstruction from Rohling et al. (2022). **(b)** $\delta^{18}O_{sw}$ reconstruction from Miller et al. (2020). **(c)** Mg / Ca-based $\delta^{18}O_{sw}$ reconstruction from North Atlantic site 607 from Dwyer and Chandler (2009) and Sosdian and Rosenthal (2009). **(d)** Mg / Ca-based $\delta^{18}O_{sw}$ reconstruction from Jakob et al. (2020). The purple line is the published $\delta^{18}O_{sw}$ reconstruction, whereas the orange line is the $\delta^{18}O_{sw}$ reconstruction based on recalibrating the Mg / Ca temperature data using Barrientos et al. (2018).

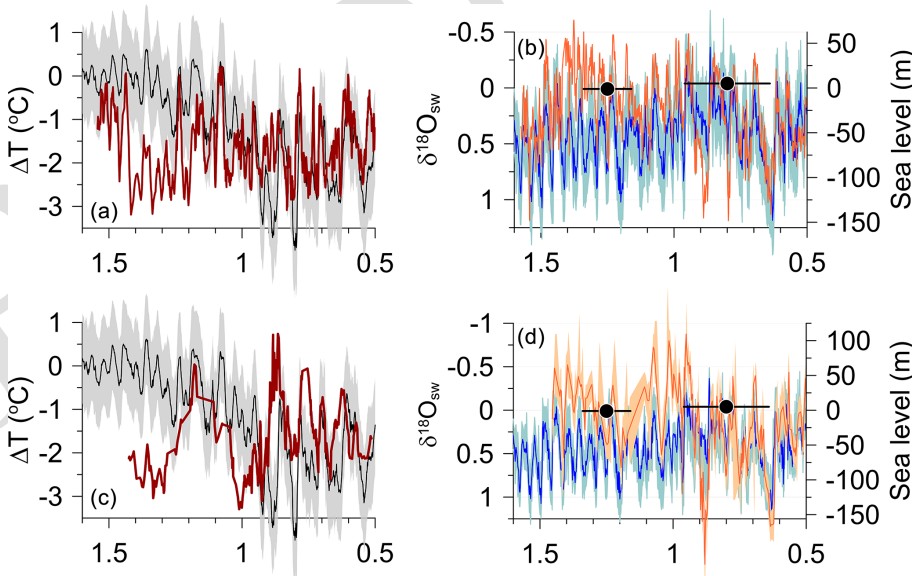

**Figure 13. (a)** Comparison of our $\Delta$MOT reconstruction (black line with $1\sigma$ uncertainty) to $\Delta$BWT reconstruction from ODP site 1123 (11-point running average shown by dark-red line) (Elderfield et al., 2012). **(b)** Comparison of our $\delta^{18}O_{sw}$-INT reconstruction (blue line with $1\sigma$ uncertainty) to the $\delta^{18}O_{sw}$ reconstruction from ODP site 1123 (orange line) (Elderfield et al., 2012). Black symbols with uncertainties are dated sea-level indicators (Dumitru et al., 2021). **(c)** Comparison of our $\Delta$MOT reconstruction (black line with $1\sigma$ uncertainty) to $\Delta$BWT reconstruction from ODP site 1208 (Ford and Raymo, 2020) (5-point running average shown by dark-red line). **(d)** Comparison of our $\delta^{18}O_{sw}$-INT reconstruction (blue line with $1\sigma$ uncertainty) to the $\delta^{18}O_{sw}$ reconstruction from ODP site 1208 (orange line with $1\sigma$ uncertainty) (Ford and Raymo, 2020). Black symbols with uncertainties are dated sea-level indicators (Dumitru et al., 2021).

tween 1.4-0.9 Ma reflect regional hydrographic changes (i.e., salinity) that were perhaps associated with the large changes in ocean circulation during the MPT (Lisiecki, 2014; Lang et al., 2016).

## 4 Processes that contribute to changes in ocean heat storage and mean ocean temperature

During time-dependent climate change, the difference between the radiative forcing at the top of the atmosphere and Earth's radiative response leads to an imbalance in the Earth's energy budget, with a positive imbalance causing the climate system to gain energy and warm and a negative imbalance causing a loss of energy that cools the climate system. Under current anthropogenic climate change, the radiative forcing has exceeded the Earth's radiative response, with 90 % of the resulting energy gain being stored in the ocean over the last few decades (von Schuckmann et al., 2023), thus strongly buffering warming of the atmosphere. The greatest ocean storage over the last century has occurred in the upper 700 m, with only weak warming at depths below 2000 m (von Schuckmann et al., 2023; Cheng et al., 2022) because of the slow transfer of energy into the ocean interior by advection, diffusion, and vertical mixing (Rugenstein et al., 2019; Gregory, 2000; Saenko et al., 2021) so that much of the ocean has not yet reached its equilibrium temperature change and HSE is only $\sim 0.1$.

Changes in ocean heat storage similarly play an important role in pacing surface temperature change on longer timescales. On orbital timescales ($10^4$–$10^5$ years), the contribution to the global energy budget from latent heat fluxes associated with large changes in land ice became comparable to changes in ocean heat storage, with each accounting for $\sim 50$ % of the increase in the internal energy of the climate system during the last deglaciation (Baggenstos et al., 2019). Changes in some combination of these two energy reservoirs indicate that the global energy budget has rarely been in balance for any extended period throughout the Plio-Pleistocene glacial–interglacial cycles (Shackleton et al., 2023).

Patterns of ocean heat storage since the late 19th century are largely associated with changes in ocean circulation that redistribute heat but do not change global heat content (Bronselaer and Zanna, 2020; Gregory et al., 2016; Cheng et al., 2022). However, observations over the last few decades and climate models show that large-scale patterns of heat storage are increasingly being determined and sustained by heat from anthropogenic surface warming that is added to the ocean interior predominantly along known water mass pathways, with this added heat dominating ocean heat storage change by 2100 (Bronselaer and Zanna, 2020; Fox-Kemper et al., 2021; Cheng et al., 2022). The patterns of OHU and storage show most warming occurring in the upper 2000 m between $60°$ S and $60°$ N, with the majority of heat uptake occurring within wind-driven subduction regions in the Southern

Ocean that ventilate the ocean interior (Kuhlbrodt and Gregory, 2012; Gregory et al., 2016), particularly in Subantarctic Mode Water and Antarctic Intermediate Water (Zanna et al., 2019; Cheng et al., 2022), a pattern that persists in equilibrium runs forced by $CO_2$ quadrupling (Fig. 14a, b) (Li et al., 2013) and in the PlioMIP2 multi-model ($n = 15$) ensemble (Fig. 14g, h) (Haywood et al., 2020). Area-averaged warming in the Pacific Ocean will be smaller than in other basins because of the lack of deepwater formation and limited formation of mode and intermediate water in the North Pacific (Cheng et al., 2022). The increase in Atlantic Ocean heat storage is projected to be nearly equivalent to that of the Pacific but, because of its smaller area, results in a significantly larger area-averaged warming (Cheng et al., 2022).

The temperature of the deep ocean ($> 2000$ m) is largely associated with deepwater formation at high latitudes. Described in broad terms, this process cools the deep ocean at a rate $v(T_u - T_d)$, where $v$ (in $s^{-1}$) is the volume rate of deepwater formation divided by the volume of the deep ocean, $T_d$ is the temperature of the water sinking at high latitude following convection, and $T_u$ is the temperature of water upwelling at lower latitude in the basin[3]. The latter temperature also represents the influence of downward mixing of heat from overlying warmer water at low latitudes. By continuously injecting cold water, the overall effect of the overturning circulation associated with NADW and AABW is to keep the deep ocean cooler than overlying intermediate-depth waters, which are ventilated by lower-latitude surface waters. A change in MOT thus occurs from some combination of changes in mid-latitude SSTs (affecting $T_u$), high-latitude SSTs or temperature of the newly formed and sinking deep water (affecting $T_d$), and high-latitude deepwater formation rate ($v$).

Projected warming in the deep North Atlantic in response to anthropogenic warming is caused by a reduction in surface heat loss in that region and in the formation of NADW in response to anthropogenic warming (Fox-Kemper et al., 2021). However, NADW does not contribute significantly to global OHU (Saenko et al., 2021), and it makes only a small contribution to changes in MOT because it ventilates only a small volume, which is currently $\sim 20$ %–30 % of the global ocean (Johnson, 2008; Khatiwala et al., 2012) that decreases during glaciations (Galbraith and de Lavergne, 2019). The correlation between AMOC and ocean heat uptake efficiency across models seems to be due to a common control, such as verti-

---

[3]If the volume $V_d$ of the ocean (in $m^3$) occupied by deep water is steady, deepwater formation at a rate $r_d$ (in $m^3 s^{-1}$) must be balanced by an equal rate of removal of deep water by upwelling and mixing with overlying water masses. The rate (in W) at which heat is removed from the deep ocean by this throughput is $r_d(T_u - T_d)C$, where $C$ is the volumetric heat capacity of seawater (in $J m^{-3} °C^{-1}$). Since the heat capacity of the deep ocean is $V_dC$ ($J °C^{-1}$), its rate of cooling (in $°C s^{-1}$) is $r_d(T_u - T_d)C/(V_dC) = v(T_u - T_d)$, where $v = r_d/V_d$, whose reciprocal $\tau = 1/v$ is the time required to renew the entire volume of the deep ocean.

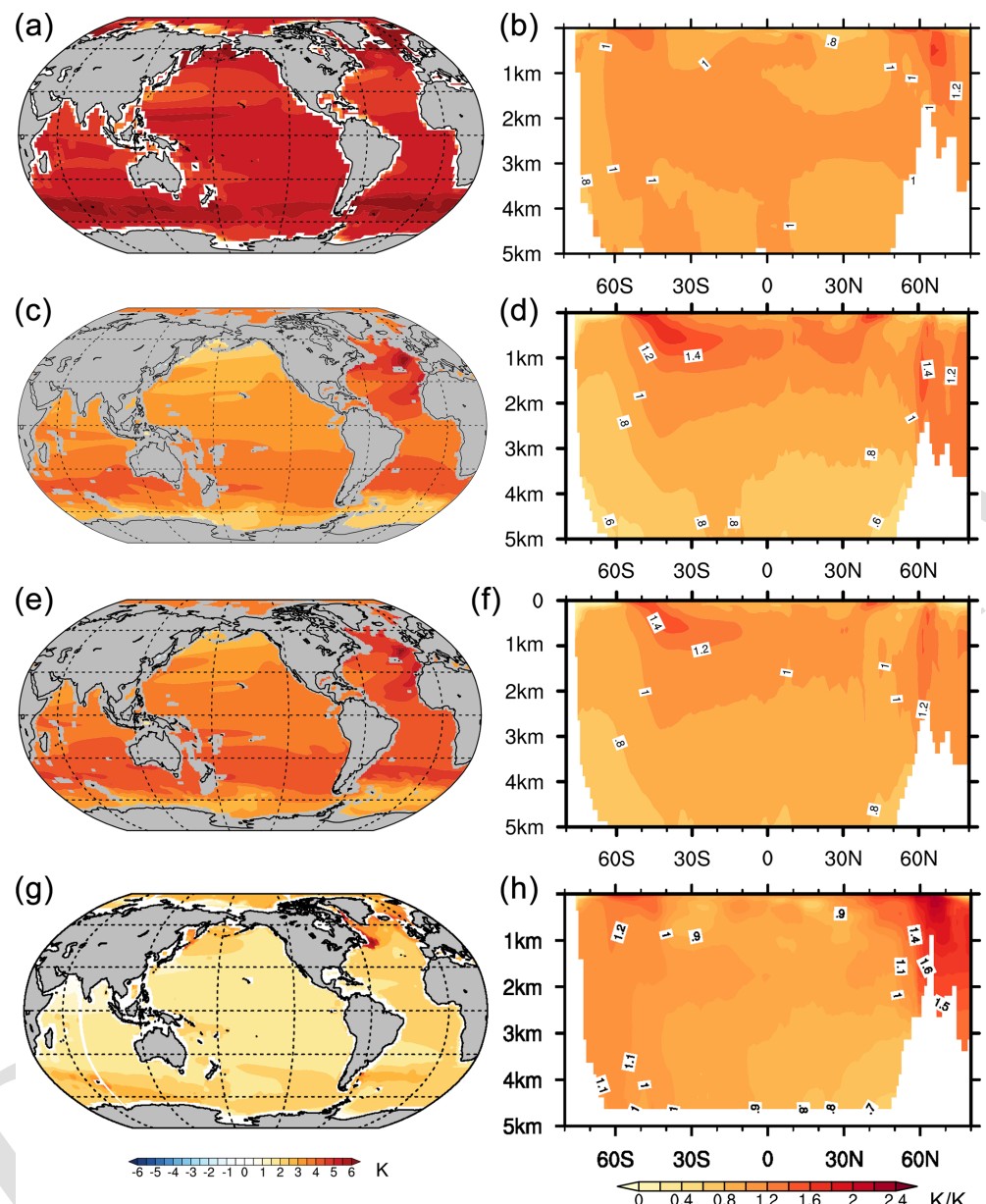

**Figure 14.** Changes in 0–2000 m mean equilibrium temperature change (left) and zonal average equilibrium ocean warming pattern normalized with $\Delta$MOT (right) for different modeling scenarios. **(a, b)** CESM 1.0.4 for abrupt $4 \times CO_2$ minus preindustrial after 5900 years, $\Delta$MOT = 5.12 K (Rugenstein et al., 2019). **(c, d)** iTRACE ICE + ORB + GHG simulation for Middle Holocene (6 ka) minus LGM, $\Delta$MOT = 3.02 K (Zhu et al., 2024). **(e, f)** Full iTRACE simulation (ICE + ORB + GHG + AMOC) for Middle Holocene (6 ka) minus LGM, $\Delta$MOT = 3.6 K (Zhu et al., 2024). **(g, h)** Model mean of PlioMIP2 simulations, $\Delta$MOT = 1.86 K (Haywood et al., 2020).

cal stratification of the global ocean, rather than to an effect of AMOC on heat uptake (Gregory et al., 2024; Newsom et al., 2023).

In contrast, AABW has a significantly larger influence on MOT than NADW because it ventilates a larger volume, which is currently $\sim 40\,\%$–$50\,\%$ of the global ocean (Johnson, 2008; Khatiwala et al., 2012), increasing to as much as $80\,\%$ during glaciations (Galbraith and de Lavergne, 2019). AABW is formed by intense heat loss and brine rejection due

to sea-ice formation, with the dense waters sinking, spreading northward to fill much of the abyss, and upwelling as they mix with overlying warm waters especially where in contact with areas of rough seafloor topography. At present, because the temperature of its source waters ($T_\mathrm{d}$ in the conceptual model above) remains near the freezing point ($\sim -1.8\,°\mathrm{C}$), the influence of AABW on abyssal temperatures has been through a reduction in the volume rate of formation ($v$ above) in response to freshening associated with increased meltwa-

ter from the Antarctic Ice Sheet (Heuzé et al., 2015; Li et al., 2023) or decreased sea-ice formation (Zhou et al., 2023), allowing more heat to diffuse or mix downwards and warm the deep ocean (Purkey et al., 2019; Johnson et al., 2024). Further global warming and associated sea-ice loss will allow AABW source waters to warm, further contributing to warming of abyssal temperatures (Fig. 14b).

This understanding of the major processes involved in OHU and storage in response to GHG emission scenarios over the course of this century (Fox-Kemper et al., 2021; Cheng et al., 2022) or on equilibrium timescales ($10^3$ years) (Fig. 14a, b, g, h) (Rugenstein et al., 2016; Li et al., 2013) contrasts with the long-standing view in paleoceanography that changes in DOT and MOT result solely from SST changes in high-latitude regions where deep water is formed (Emiliani, 1954; Zachos et al., 2001; Hansen et al., 2023; Evans et al., 2024; Westerhold et al., 2020; Bereiter et al., 2018; Rohling et al., 2022; Hansen et al., 2013). Changes in source water temperature may indeed cause changes in deep-water temperature, for example, during substantially warmer climates without Antarctic sea ice (Goudsmit-Harzevoort et al., 2023; Evans et al., 2024) when AABW could form during Antarctic winter solely by heat loss to the atmosphere without brine rejection, like NADW in the present climate. However, the general notion of a sole control of MOT by SSTs at sites of deepwater formation (affecting $T_d$) should not be applied regardless of climate state because it neglects the roles of the rate ($v$) of deepwater formation and the contribution from OHU in mid-latitudes, both of which can also affect ocean heat storage (Fig. 14a, b) and thus contribute to MOT.

Zhu et al. (2024) used results from the iTRACE simulation of the last deglaciation to assess the role of the primary forcings of Plio-Pleistocene climate change on MOT change. The iTRACE ORB + ICE + GHG simulation includes most of the key forcings during the Plio-Pleistocene ice-age cycles. Orbital forcing has little direct influence on GMSST, leaving ice sheets as the primary forcing that modulates the MOT response to GHG forcing. The effect of lowering of Northern Hemisphere ice sheets as they retreat induces surface warming that is advected downwind to the North Atlantic and North Pacific, further enhancing SST warming (Fig. 15) and OHU through ventilation of intermediate waters at 45° N (Fig. 14d). Ice-sheet forcing in the ORB + ICE + GHG simulation thus significantly enhances intermediate-depth warming at that is otherwise largely occurring through wind-driven ventilation in the Southern Ocean (Fig. 14c).

Zhu et al. (2024) found that high-latitude source waters where deep water is formed are largely covered by sea ice, resulting in their temperature remaining around the freezing point throughout much of the deglaciation. The presence of sea ice results in peak SST warming during deglaciation occurring in mid-to-subpolar latitudes (e.g., Fig. 15) as opposed to peak surface air temperature warming occurring at high latitudes through polar amplification (Zhu et al., 2024).

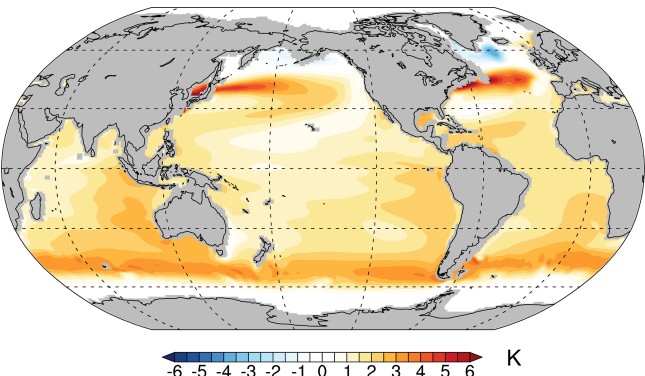

**Figure 15.** iTRACE ICE-only (ICE run) simulation of ocean surface temperature change (upper 10 m) for the Middle Holocene (6 ka) minus LGM.

Strong ventilation regions remained co-located with peak SST warming throughout the deglaciation, particularly in the mid-latitudes of the Southern Ocean associated with ventilation of Antarctic Intermediate Water. Figure 14d shows that this was the primary pathway of warming the global ocean, with a negligible warming contribution from reduced AABW formation (Zhu et al., 2024). During the deglaciation, the iTRACE ORB + ICE + GHG simulation shows that MOT warming lags SST warming by several thousand years, reflecting the timescale of warming the ocean interior by ventilation of intermediate waters and resulting in an average transient HSE ∼ 0.5 that disagrees with proxy records (Zhu et al., 2024). However, this simulation shows that the lag is short enough to allow MOT warming to reach equilibrium during the present interglaciation and HSE reaches ∼ 1.

The full iTRACE simulation includes meltwater forcing (MWF) associated with Heinrich event 1 and the Younger Dryas (He et al., 2021; Gu et al., 2020) and thus captures MOT changes during times of millennial-scale AMOC variability. The strong AMOC weakening in response to MWF causes northward heat transport to decrease, resulting in the characteristic SST bipolar seesaw pattern of Northern Hemisphere cooling and Southern Hemisphere warming. GMSSTs do not change substantially, however, because the effects in both hemispheres on SST nearly cancel each other. In contrast, a suppression of NADW production and reduction of the AMOC generates a subsurface warming that extends to intermediate depths over much of the global ocean (Fig. 14e) and warmed abyssal waters through circulation and mixing processes (Fig. 14f). The MWF causes the subsurface ocean warming that eventually occurs from orbital forcing and land ice on temperature to occur more rapidly. Thus it eliminates the unrealistic lag of MOT behind GMSST, resulting in HSE ≥ 1 throughout the deglaciation, in agreement with proxy records (Zhu et al., 2024).

## 5 A general hypothesis for the increase in ocean heat storage efficiency during the Middle Pleistocene Transition

Based on the understanding of controls on OHU and MOT outlined in Sect. 4, we develop a working hypothesis for the reconstructed increase in HSE that occurred during the MPT. The main premise of our simple conceptual model is based on the ocean being comprised of an upper ocean heat reservoir (herein $R_{<2000}$) that extends from roughly 50° S to 50° N and to a depth of about 2000 m and has a volume that in PlioMIP2 is 55 % of the global ocean ($f_{<2000} = 0.55$) and in iTRACE is 43 % of the global ocean ($f_{<2000} = 0.43$), with the heat content and temperature being largely determined by ventilation of mid-latitude surface waters (Fig. 14). The remainder of the ocean (herein $R_{>2000}$), the deeper ocean heat reservoir, is largely below 2000 m, is connected to the surface at latitudes of $> 50°$ S and $> 50°$ N, and represents 45 % (PlioMIP2) and 57 % (iTRACE) of the global ocean volume, with the heat content and temperature being largely determined by high-latitude deepwater formation (some combination of $T_d$ and $v$). In this simple model, $\Delta$MOT is equal to the mean of the changes in temperatures of the upper reservoir $R_{<2000}(\Delta T_{<2000})$ and deeper reservoir $R_{>2000}(\Delta T_{>2000})$ weighted by their relative ocean volumes ($\Delta$MOT $= f_{<2000} \times \Delta T_{<2000} + (1 - f_{<2000}) \times \Delta T_{>2000}$).

We derive parameters from climate models that identify the relation of the temperature of the upper reservoir to $\Delta$GMSST and then calculate the temperature of the two reservoirs prior to the MPT ($> 1.5$ Ma) and since 1.5 Ma. For $> 1.5$ Ma, we use the PlioMIP2 results (Haywood et al., 2020) to calculate that the average temperature change of the upper reservoir $\Delta T_{<2000}$ is 80 % that of $\Delta$GMSST, corresponding to a multiplying factor $s = \Delta T_{<2000}/\Delta$GMSST of 0.8. For the time since 1.5 Ma, we use the full iTRACE results (Zhu et al., 2024) to calculate that the average temperature change of the upper reservoir $\Delta T_{<2000}$ is 16 % greater than $\Delta$GMSST, corresponding to a multiplying factor $s = \Delta T_{<2000}/\Delta$GMSST of 1.16. As a simple scaling analysis, we use these model parameters to derive upper reservoir $\Delta T_{<2000}$ from the $\Delta$GMSST reconstruction (Fig. 16a) by multiplying it with a factor $s = 0.8$ for $> 1.5$ Ma and $s = 1.16$ for $< 1.5$ Ma, including $\sigma_{\Delta T<2000} = s \times \sigma_{\Delta GMSST}$ (Fig. 16c). The temperature change of the deep reservoir $\Delta T_{>2000}$ for these two time intervals is readily derived from the equation for MOT, with its uncertainty being the square root of the sum of squares of the individual uncertainties ($\sigma_{>2000} = (\sigma_{\Delta MOT}/(1 - f_{<2000}))^2 + (f_{<2000} \times s \times \sigma_{\Delta GMSST})^2$) TS7 (Fig. 16d). For the interval since 1.5 Ma, we assessed the sensitivity of upper reservoir $\Delta T_{>2000}$ to different values of $f_{<2000}$ (in the range 0.4–0.6) and of $s$ (1.11–1.54), with all results falling well within the $1\sigma$ uncertainty of $\Delta T_{>2000}$ (not shown).

Albeit highly idealized, this simple analysis suggests that, for the period from 4.5 Ma until the start of the MPT around 1.5 Ma, $\Delta T_{<2000}$ based on the nominal PlioMIP2 parameters ($s = 0.8$, $f_{<2000} = 0.55$) (Fig. 16c) accounts for nearly all of $\Delta$MOT (Fig. 16b) (which in this time window has an HSE of 0.5), leaving only a small decrease in $\Delta T_{>2000}$ (Fig. 16d) and thus providing an explanation for HSE being $\sim 0.5$. In other words, before the MPT, nearly all of $\Delta$MOT is occurring in the upper reservoir, which is ventilated by the wind-driven circulation, and cools along with $\Delta$GMSST. The global cooling trend throughout this period is assumed to be a response to declining $CO_2$ (Clark et al., 2024), but the cause of it does not affect our argument. Meanwhile, little change is occurring in the high-latitude deepwater formation rate ($v$) and sinking water temperature ($T_d$) and thus in deep reservoir temperature ($\Delta T_{>2000}$).

The lack of substantial deep reservoir $\Delta T_{>2000}$ change on orbital ($10^4$–$10^5$ years) and geological ($10^6$ years) timescales between 4.5–1.5 Ma suggests relatively stable and constant AABW formation, with a subsequent decrease around 1.5 Ma in long-term $\Delta T_{>2000}$ and a rise in its variability, suggesting that significant changes in AABW formation had begun. Two lines of evidence support this scenario. First, multi-model results from PLIOMIP2 found that Southern Ocean $\Delta$SSTs for the KM5c interglaciation at 3.205 Ma were $2.8 \pm 1.3$ °C (Weiffenbach et al., 2024) (Fig. 16e). These warmer SSTs, combined with a simulated increase in precipitation and decrease in sea-ice cover, resulted in a strongly stratified Southern Ocean with relatively uniform warming of 1.5–2.5 °C throughout much of the water column below the low-density surface layer. In 9 of the 15 PLIOMIP2 models, this increase in stratification led to a decrease in AABW formation, with Weiffenbach et al. (2024) noting that four of the other six models also have greater stratification, but the AABW response may be modulated by interactions with a stronger AMOC in those models.

To examine whether these mid-Pliocene boundary conditions extended into the Pleistocene, we use the SH extratropical ($> 30°$ S) $\Delta$SST stack from Clark et al. (2024) as a proxy for Southern Ocean SSTs. (Note that this stack only extends to 4 Ma because of the limited number of older records available, and so our analysis here only covers the last 4 Myr.) This inference is supported by the good agreement with the $\Delta$SST reconstruction for the Southern Ocean derived from deuterium excess from the Dome Fuji ice core for the last 0.7 Myr (Uemura et al., 2018) and the PLIOMIP2 multi-model mean Southern Ocean $\Delta$SST of $2.8 \pm 1.3$ °C at 3.205 Ma (Weiffenbach et al., 2024) compared to the stack's $2.4 \pm 1.3$ °C (Clark et al., 2024) (Fig. 16e). At the same time, the PLIOMIP2 models find a linear relation between Southern Ocean $\Delta$SST and $\Delta$sea-ice area in percentages relative to preindustrial times where $\Delta$sea-ice area $= -12.4$ % °C$^{-1} \times \Delta$SST (Weiffenbach et al., 2024), which we apply to our $\Delta$SST stack to derive relative changes in sea-ice area over the last 4 Myr (Fig. 16e).

These results suggest that the highly stratified Southern Ocean found in PLIOMIP2 simulations at 3.2 Ma due

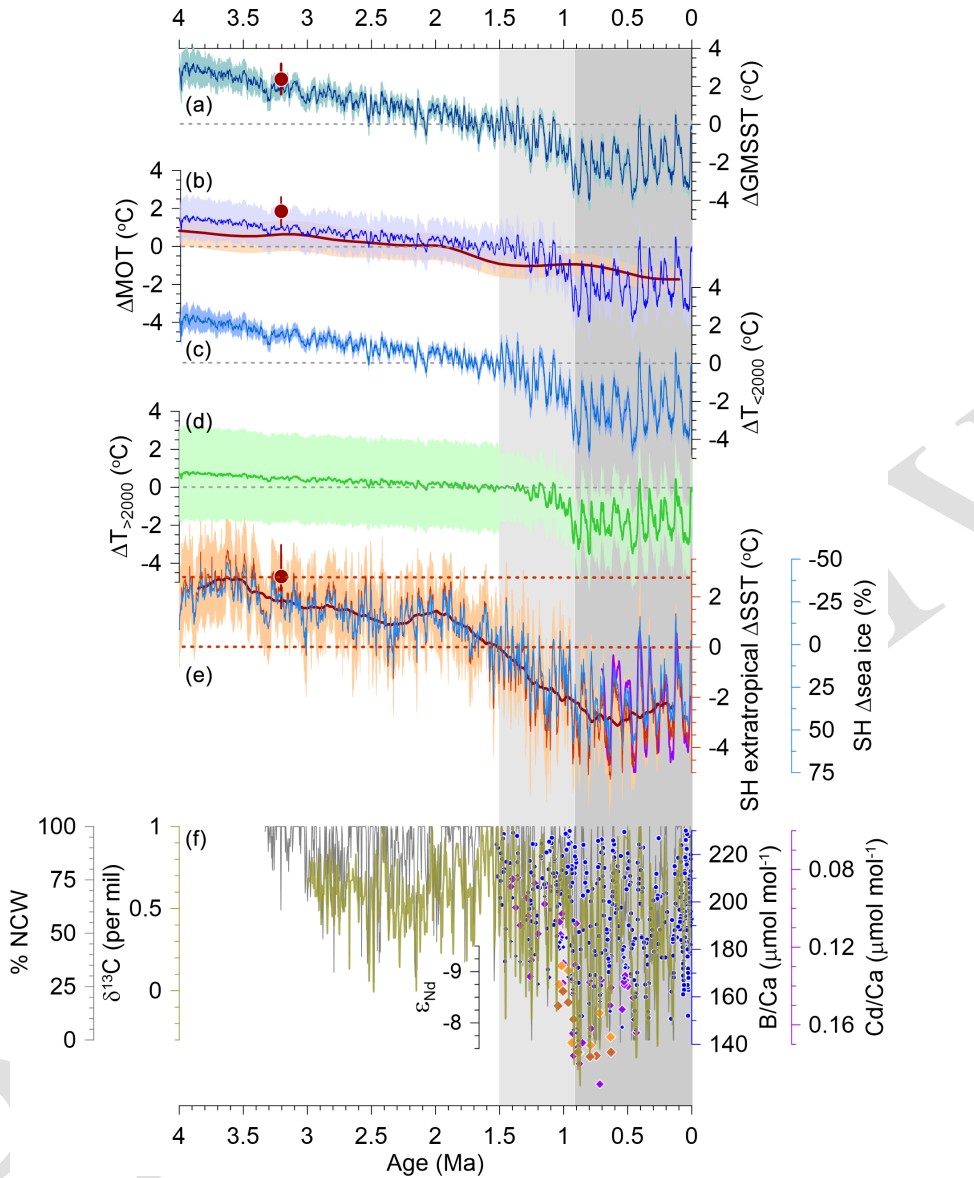

**Figure 16. (a)** Global mean sea surface temperature change from preindustrial (PI) (ΔGMSST) (dark-blue line with 1σ uncertainty) (Clark et al., 2024). Red symbol with 1σ uncertainty is the model mean ΔGMSST during the KM5c time slice at 3.205 Ma from PlioMIP2 (Haywood et al., 2020). **(b)** Mean ocean temperature change from PI (ΔMOT) (blue line with 1σ uncertainty) as derived in this paper. Also shown is the smoothed ΔBWT reconstruction from Cramer et al. (2011) using their Eq. (7b) (brick-red line with 90% confidence interval). The red symbol with 1σ uncertainty is the model mean MOT during the KM5c time slice at 3.205 Ma from PlioMIP2 (Haywood et al., 2020). **(c)** $\Delta T$ for the $R_{<2000}$ area of ocean (43% of global volume) (lighter blue line with 1σ uncertainty) that is on average 16% greater than ΔGMSST ($\Delta T\_R_{<2000} = \Delta GMSST \times 1.16$) (Zhu et al., 2024). **(d)** $\Delta T$ for the $R_{>2000}$ area of ocean (57% of global volume) (green line with 1σ uncertainty) derived by $\Delta T\_R_{>2000} = (\Delta MOT - 0.43 \times \Delta T\_R_{<2000})/0.57$. **(e)** Southern Hemisphere extratropical ΔSST stack (brick-red line with 1σ uncertainty and 201 kyr running average in dark red) (Clark et al., 2024), Southern Ocean Δsea-ice extent derived from the relation to the Southern Hemisphere extratropical ΔSST stack established by PLIOMIP2 models (Weiffenbach et al., 2024) (light-blue line), ΔSST reconstruction for the Southern Ocean for the last 0.7 Myr derived from deuterium excess from the Dome Fuji Antarctic ice core (purple = 25 kyr running average) (Uemura et al., 2018), and PlioMIP2 simulated multi-model mean ΔSST reconstruction for the Southern Ocean during the KM5c time slice at 3.205 Ma shown by the red symbol and 1σ uncertainty (2.8 ± 1.3 °C) (Weiffenbach et al., 2024). Two dashed red horizontal lines correspond to PLIOMIP2 ΔSST at 3.205 Ma (2.8 °C) and at PI (0 °C). **(f)** $\delta^{13}$C stack of mid-to-deep Atlantic cores (green) (Lisiecki, 2014), percent of northern component waters (NCWs) (gray) (Lang et al., 2016), $\varepsilon_{Nd}$ data from South Atlantic sites ODP 1088 (light-orange diamonds) and 1090 (red-brown diamonds) (Pena and Goldstein, 2014), and Cd / Ca (purple diamonds) and B / Ca data from North Atlantic cores CHN82-24-23PC and DSDP 607 (blue circles) (Sosdian et al., 2018; Lear et al., 2016) and South Atlantic site ODP 1267 (blue diamonds) (Farmer et al., 2019).

to warm SSTs and reduced sea-ice extent persisted until $\sim 1.5$ Ma. Prior to this, SSTs and sea-ice extent spent 90 % of the time above and below, respectively, their preindustrial values. Apparently, SSTs in the AABW formation regions did not decline sufficiently during this interval to affect $\Delta T_{>2000}$ and MOT substantially by lowering $T_d$. The start of the MPT at 1.5 Ma saw a significant change in the influences on AABW, with SSTs spending increasingly more time below preindustrial levels and sea-ice extent spending more time above preindustrial values ($\sim 85$ % during the MPT, $\sim 95$ % since 0.9 Ma). This might also be related to the proposed decoupling of Southern Ocean vertical mixing and Southern Ocean SST prior to the MPT (Köhler and Bintanja, 2008). We thus conclude that it was the persistence of a highly stratified Southern Ocean that caused a decrease in AABW formation rate and persistently warmer $T_d$ than present until $\sim 1.5$ Ma, when the gradual decay of stratification and increase in sea-ice extent and variability then enhanced conditions for AABW formation.

Second, deep-ocean water masses show changes that are consistent with changes in AABW formation inferred from our simple model; i.e., $v$ increased at that time. Starting at $\sim 1.5$ Ma, there was an increasing frequency of southern component waters (e.g., AABW) (Lang et al., 2016), which are depleted in $\delta^{13}C$ (Lisiecki, 2014), recording a growing influence of AABW at the expense of NADW in the Atlantic Ocean (Fig. 16f). This was followed by a further step-change increase in the relative share of $\delta^{13}C$-depleted AABW during glacial climates around 0.9 Ma, as also indicated by a rapid increase in $\varepsilon_{Nd}$ values (Pena and Goldstein, 2014) and an increase in nutrient content and a decrease in carbonate ion saturation as shown in the Cd / Ca and B / Ca records, respectively (Lear et al., 2016; Sosdian et al., 2018; Farmer et al., 2019) (Fig. 16f).

After the MPT, $\Delta$MOT variability was greater than before, exceeding the contribution from upper reservoir $\Delta T_{<2000}$. During the glacial cycles of the last 0.8 Myr, proxy records suggest that $v$ has varied along with $\Delta$GMSST (Fig. 16f) (Clark et al., 2024). Without requiring any long-term change in $T_d$, which has remained near freezing, this can explain synchronized variations in $\Delta T_{<2000}$ and $\Delta T_{>2000}$, leading finally to an HSE of $\sim 1$ during this period.

## 6   Summary

When compared to a reconstruction of $\Delta$GMSST over the last 4.5 Myr, high-fidelity proxies of deep ($> 200$ m) ocean temperature change show good agreement in orbital-scale amplitude and long-term trend over the last 0.7 Ma, but their long-term trends are $\sim 50$ % of long-term $\Delta$GMSST beyond 1.5 Ma, suggesting an increase in HSE from $\sim 0.5$ to $\sim 1$ during the MPT (1.5–0.9 Ma). This increase is further supported when assuming that HSE was 1 throughout the last 4.5 Myr and applying this temperature history (as $\delta^{18}O_T$)

to isolate the seawater component ($\delta^{18}O_{sw}$) of a probabilistic global $\delta^{18}O_b$ stack (Prob-stack). Under this scenario, Pliocene $\delta^{18}O_{sw}$ values are 0.3‰ to 0.5‰ despite robust evidence for higher-than-present Pliocene sea levels that require values in $\delta^{18}O_{sw}$ to be smaller than 0‰, suggesting that too much of the $\delta^{18}O_b$ signal is being removed by the $\delta^{18}O_T$ component using an HSE of 1. Applying our $\Delta$MOT reconstruction where HSE increases from 0.5 to 1 across the MPT results in Early Pleistocene and Pliocene $\delta^{18}O_{sw}$ values (0‰ to $-0.1$‰) that continue to be more positive than multiple data constraints. While a further decrease in HSE to 0.1 would result in average Pliocene $\delta^{18}O_{sw}$ values of $-0.2$‰, which could explain higher sea levels at that time, such a low HSE is ruled out by proxy-based bottom water temperature reconstructions. We therefore adopt the hypothesis that there has been a diagenetic overprint on $\delta^{18}O_b$ records that average to a long-term increase of between 0.05‰ Myr$^{-1}$ and 0.12‰ Myr$^{-1}$ which, when removed from the Prob-stack, results in Pliocene $\delta^{18}O_{sw}$ values of $-0.1$‰ to $-0.4$‰ that are consistent with sea-level highstands of 20–25 m above present.

To explain the increase in HSE across the MPT, we develop a simple conceptual model that considers the ocean as being comprised of an upper non-polar ocean reservoir with the temperature being largely determined by ventilation of mid-latitude surface waters and a deeper ocean reservoir whose temperature is largely determined by high-latitude deepwater formation. Using results from a transient simulation of the last deglaciation with a global climate model, we develop a simple scaling analysis to derive upper reservoir $\Delta T$ from the $\Delta$GMSST reconstruction, which is then subtracted from $\Delta$MOT to derive deep reservoir $\Delta T$. This analysis suggests that before the MPT, nearly all of $\Delta$MOT is occurring in the upper reservoir through changes in wind-driven ventilation and little is occurring in the deep reservoir from changes in deepwater formation, resulting in HSE being $\sim 0.5$. Around 1.5 Ma, the amplitude of $\Delta$MOT variability begins to increase and exceeds the contribution from upper reservoir $\Delta T$, thus requiring an increasing contribution of lower reservoir $\Delta T$ to $\Delta$MOT through an increase in deepwater formation that leads to an HSE of $\sim 1$ over the last 0.8 Myr. We attribute these changes in deepwater formation to long-term cooling which caused a transition starting $\sim 1.5$ Ma from a highly stratified Southern Ocean due to warm SSTs and reduced sea-ice extent to colder SSTs with a significant increase in sea-ice extent and more vertical exchange of water masses.

**Code and data availability.** Our $\Delta$MOT and $\delta18O_{sw}$ data are available in the Supplement. R code that implements the regression analysis between $\Delta$MOT and $\Delta$GMSST can be found at Zenodo (https://doi.org/10.5281/zenodo.14759006, Bartlein, 2025) or in the GitHub repository at (https://github.com/pjbartlein/MOTvsSST, last access: 2025 TS8).

**Supplement.** The supplement related to this article is available online at [the link will be implemented upon publication].

**Author contributions.** Conceptualization: PUC and JDS. Methodology: PUC, JDS, YR, CZ, DPS, and PK. Investigation: PUC, JDS, YR, CZ, JMG, PK, ZL, DPS, and PJB. Writing – original draft: PUC, JDS, and YR. Writing – review and editing: PUC, JDS, YR, JMG, PK, CZ, ZL, PJB, and DPS.

**Competing interests.** The contact author has declared that none of the authors has any competing interests.

**Disclaimer.** Publisher's note: Copernicus Publications remains neutral with regard to jurisdictional claims made in the text, published maps, institutional affiliations, or any other geographical representation in this paper. While Copernicus Publications makes every effort to include appropriate place names, the final responsibility lies with the authors.

**Acknowledgements.** We thank Lorraine Lisiecki and Thomas Stocker for their constructive reviews. We thank the paleoclimate and paleoceanographic communities for making their data sets widely available; the National Centers for Environmental Information of NOAA and the World Data Center PANGAEA for archiving data; and Chris Brierley, Julia Tindall, and Julia E. Weiffenbach for providing climate model data. This publication contributed to Beyond EPICA, a project of the European Union's Horizon 2020 research and innovation program (Oldest Ice Core). This is Beyond EPICA publication number 43. Peter U. Clark is funded by National Science Foundation OPP-2103032, Yair Rosenthal is supported by National Science Foundation OCE-1834208, and Zhengyu Liu is supported by National Science Foundation OCE-1810681.

**Financial support.** This research has been supported by the National Science Foundation (grant nos. OPP-2103032, OCE-1834208, and OCE-1810681).

**Review statement.** This paper was edited by Denis-Didier Rousseau and reviewed by Lorraine Lisiecki and Thomas Stocker.

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

## Remarks from the language copy-editor

CE1    Please give an explanation of why this needs to be changed. We have to ask the handling editor for approval. Thanks.

CE2    Please give an explanation of why this needs to be changed. We have to ask the handling editor for approval. Thanks.

CE3    Please give an explanation of why this needs to be changed. We have to ask the handling editor for approval. Thanks.

CE4    Please note that it's our standard to differentiate between the singular and plural forms of abbreviations. As the definitions here are plural, the abbreviations are also plural. This applies to other similar instances of plural abbreviations.

CE5    Should this instance also be "modern values"?

CE6    The initial sentence had to be rephrased; please check and confirm or offer an alternative.

## Remarks from the typesetter

TS1    Please note that not all changes could be inserted. The proofreading process is reserved for responses to the publisher remarks and, if applicable, spotting clear mistakes included during the production process. Updates to the manuscript content cannot be accepted and a further proofreading round is not foreseen (https://publications.copernicus.org/for_authors/proofreading_guidelines.html).

TS2    Please note that it is our standard to use skinny spaces for ratios. Therefore, the requested changes could not be inserted.

TS3    Due to the requested changes in the figures, we have to forward your requests to the handling editor for approval. To explain the corrections needed to the editor, please send me the reason why these corrections are necessary. Please note that the status of your paper will be changed to "Post-review adjustments" until the editor has made their decision. We will keep you informed via email.

TS4    Please give an explanation of why this needs to be changed. We have to ask the handling editor for approval. Thanks.

TS5    Please give an explanation of why this needs to be changed. We have to ask the handling editor for approval. Thanks.

TS6    Please give an explanation of why this (and the other instances) needs to be changed. We have to ask the handling editor for approval. Thanks.

TS7    Please check subscripts.

TS8    Please provide the day and at least the month.

TS9    Please provide date of last access.

TS10    Please confirm.

TS11    Please provide day and month.