# Peer review of "Mean ocean temperature change and decomposition of the benthic $\delta^{18}$ O record over the past 4.5 million years"

_EGUsphere, 2024_

## Author Comment (AC1)

This manuscript is likely to be of great interest to the paleoceanography community because it makes significant progress in finding a self-consistent decomposition of global mean benthic d18O into temperature and seawater (ice volume) components in a way which is consistent with independent estimates of global mean sea surface temperature (GMSST) and sea level constraints. Overall, it is well written and well supported by evidence. However, the manuscript could be significantly improved with some additional clarification.

Thank you for this positive assessment and very helpful comments which have improved the manuscript.

We first point out that, at the suggestion of Reviewer 2, we downloaded the 15 model results from the PLIOMIP2 experiment (e.g., Haywood et al., 2020, *Climate of the Past*) to obtain ΔGMSST and ΔMOT for each model run. We added these to the model results shown in Fig. 2A and reassessed the HSE using several statistical models. As explained in the revised text, two statistical models (LOESS and segmented regression with two breakpoints) provide equivalent fits to the data that are superior to the linear regression used in the original Fig. 2A. These model results now suggest that HSE is 1 for ΔGMSST <0°C, 0.6 for ΔGMSST 0°C to 5°C, and 1.2 for ΔGMSST >5°C. These results are thus consistent with what we derived from the proxy data for the last 4.5 Ma (i.e.,), providing two independent lines of evidence for an increase in HSE during the MPT from ~0.5 to 1. These new results have been incorporated into the revision.

Major points:
1. The calculations of mean ocean temperature (MOT) change relies on a transition in the ocean heat storage efficiency (HSE) from ~0.5 before the MPT to ~1 after the MPT. While the need for such a transition is well justified by comparison with BWT measurements, the available proxy data before the MPT (particularly in the Pacific) are quite sparse with large scatter and uncertainties. Although the authors appropriately provide a large uncertainty estimate for HSE, they provide calculations for the decomposition of d18Osw using only one scenario, in which HSE changes linearly between 1.5-0.9 Ma. It would be enormously helpful for the interpretation of the d18Osw estimate if the authors also provided the d18Osw results of a few sensitivity tests in which the timing and amplitude of HSE change are varied within the range consistent with BWT estimates.

This is an excellent suggestion. Some of this can be inferred from existing information in the paper. For example, Figure 6B shows ΔMOT reconstructions based on HSE = 0.7 and HSE = 0.3, which closely encompass the 1σ uncertainty on our ΔMOT reconstruction based on HSE = 0.5, suggesting that $\delta^{18}O_{sw}$ based on HSE = 0.3 and 0.7 will similarly fall within the uncertainty of our $\delta^{18}O_{sw}$ based on HSE = 0.5. Figures 9A and 9B similarly indicate the sensitivity of $\delta^{18}O_{sw}$ based on HSE = 1 and 0.5.

To address the reviewer's comment more thoroughly, we have added four panels to Figure 9 (see below). Panels E and F address the amplitude question. Panel E (reproducing Figure 6B) shows that ΔMOT reconstructions based on HSE = 0.7 and HSE = 0.3 fall within the 1σ uncertainty on our ΔMOT reconstruction based on HSE = 0.5. Panel F shows our high-resolution $\delta^{18}O_{sw}$ (violet) with 1σ uncertainty compared to long-term (401-kyr running average) $\delta^{18}O_{sw}$ for the three HSE scenarios. The differences during the early Pleistocene are small (<0.1 per mil), and the highresolution $\delta^{18}O_{sw}$ for the two bracketing HSE scenarios fall within the $1\sigma$ uncertainty of high-resolution $\delta^{18}O_{sw}$ based on 0.5-1 HSE scenario.

Panels G and H address the timing question by comparing our preferred scenario (HSE increased from 0.5 to 1 between 1.5 Ma and 0.9 Ma) to one scenario where increase occurred more rapidly (1.2-0.9 Ma) and another where it increased more gradually (1.7-0.7 Ma). Panels G and H show that the differences in $\Delta MOT$ and $\delta^{18}O_{sw}$, respectively, are negligible.

We thus conclude that our main findings regarding $\delta^{18}O_{sw}$ are robust to the range of $\Delta MOT$ suggested by the models and data.

[Figure]

2. Similarly, the timing of the hypothesized diagenetic alteration of benthic d18O is not well constrained by proxy data. Although Raymo et al (2018) proposed a simple linear trend for this effect, one might alternatively hypothesize that the effect would covary with MOT or BWT change if the mechanism responsible for the effect is the cooling of BWT. Because the manuscript estimates that MOT cools most dramatically during the MPT, it would be informative to also show the results of a sensitivity test in which the rate of diagenesis is greater for d18O immediately preceding the MPT (keeping the same estimated total diagenetic contribution at 3 Ma).

We do show sensitivity to different linear trends, ranging from 0.05 to 0.12 ‰ Myr[-1] (Fig. 9C). These yield small differences for the past 1 to 2 Myr.

We have now also followed the reviewer's suggestion and applied a diagenetic correction of similar overall magnitude but that tracks the MOT reconstruction through time – shown in the figure below. The faster rate of change across the MPT slightly increases the trend in d18Osw over this interval (more depleted before 1 Ma, more enriched after 1 Ma), but the changes are small, ≤ 0.1‰.

[Figure]

3. These two sensitivity tests would be particularly helpful for interpreting the unexpected observation that smoothed d18Osw and glacial maxima d18Osw at ~1.5 Ma are similar to (or possibly more enriched than) the d18Osw of the Late Pleistocene. It's important to clarify whether this finding is relatively robust to the specified timing and amplitude of HSE change and d18O diagenesis, neither of which is well constrained by the available proxy data.

We think our responses show that our findings are robust. In any event, we want to emphasize that our sensitivity results in Figure 9A-9C show that one obtains unrealistic $\delta^{18}O_{sw}$ values in the Pliocene and early Pleistocene without accounting for a change in HSE suggested by data and models and a change in some long-term control such as diagenesis or the carbonate ion effect.

4. An additional surprising result is the relative amplitudes of orbital-scale MOT variability and orbital-scale d18Osw variability in the pre-MPT time period. The pre-MPT MOT record contains very weak glacial-interglacial change compared to relatively large amplitude d18Osw changes from 2.6-1.5 Ma. The authors should add some discussion of the reliability of the amplitudes of the orbital-scale signal in GMSST change and MOT change. Are the resolution and age uncertainty of the SST records sufficient to accurately estimate orbital-scale changes in GMSST and, thus, its application to estimating orbital responses in MOT and d18Osw?

We addressed these issues at length in the Supplementary Material of Clark et al. (2024, Science) (see p. 5-9, Figs. S4-S6), and reached the following conclusions.
(1) Our assessment of resolution on the variability of our SST stack suggests minimal loss of the 100- and 41-kyr signals and that that despite some loss of the 23-kyr signal, it remains readily detectable.
(2) Our assessment of age model uncertainties suggests there is minimal preferential signal loss due to age misalignments.
(3) We used three different statistical models to assess whether the trends in the standard deviations of our global stack and individual-record averages differ from one another, with our results suggesting that the trends parallel one another and would be interpreted similarly in terms of the evolution of variability over the past 4 Ma.
(4) Finally, we compared our stack to several other composite reconstructions and found that all reconstructions show a gradual increase in variability over the Pleistocene similar to the stack.

We address the reviewer's comment by adding the following to our revised text:
We note that Clark et al. (2024) found little loss of variability in the ΔGMSST reconstruction due to age uncertainties and resolutions of individual SST records used in the reconstruction.

5. In Figure 13, the authors present a very interesting comparison of BWT and d18Osw estimates from two Pacific cores and their global compilation estimates. They make the compelling argument that the estimates from the two cores are unlikely to provide reliable global estimates because they imply that sea level would need to be ~50 m higher than PI for significant amounts of time between 1.4-1 Ma, suggesting that these sites may be affected by local salinity changes. Could the authors slightly expand upon this idea to explain how the locations of those Pacific cores could have significantly different bottom properties than the rest of the deep Pacific?

Unfortunately, having just the two widely spaced records cannot address this question beyond our statement that they "reflect regional hydrographic changes (i.e., salinity) that were perhaps associated with the large changes in ocean circulation during the MPT." We can speculate that there might have been a different $\delta^{18}$O-salinity relationship, which is expected if $\delta^{18}$O of Antarctic ice was not as negative or there may be a problem with the Mg/Ca data. If these records represent the whole Pacific, we need to have saltier water elsewhere to keep the salt and O isotope budget of the ocean. In any event, at this point it is hard to tell, which is why we are using our approach.

6. I really appreciated the section of the paper using model results to explore the mechanisms responsible for scaling between MOT and GMSST and why it might differ before the MPT. However, one question I have is about the authors' apparent conclusion that AABW's contribution to MOT was constant (and approximately equal to pre-industrial) from 4-1.5 Ma (Figure 16D). How can this be consistent with the PlioMIP2 findings that the deep Southern Ocean was 1.5-2.5 C warmer than pre-industrial

As noted above, we have now downloaded the PLIOMIP2 data which allows us to use their constraints on the volume of the two ocean heat reservoirs and the relationship of the temperature of the upper reservoir to GMSST. Using these improved constraints, we now find a ~1°C decrease of the temperature of our deeper reservoir ($\Delta T_{>2000}$) from 4-1.5 Ma which, with the $\pm 2$°C uncertainty of our approach, can readily accommodate the warmer Pliocene deep Southern Ocean. More importantly, however, we emphasize that our $\Delta T_{>2000}$ is for the entire ocean >2000 m, not just the Southern Ocean. Our new analysis of PLIOMIP2 $\Delta$MOT (now included in Figure 14) shows that much of the deep ocean warming is less than the deep Southern Ocean, as is expected as AABW moves northward.

and that increased stratification caused decreased AABW formation?

We refer to the PlioMIP2 findings in Weiffenbach et al. (2024) that increased surface stratification (because of warmer SSTs and less sea ice) caused a decrease in AABW formation (which leads to warming):
*We thus conclude that it was the persistence of a highly stratified Southern Ocean that caused a smaller AABW formation rate and persistently warmer Td than present until ~1.5 Ma, when the gradual decay of stratification and increase in sea-ice extent and variability then enhanced conditions for AABW formation.*

Minor points:
Line 546: The statement that 1123 records large ice sheets pre-MPT is unclear because most of the pre-MPT d18Osw record is significantly lighter than the post-MPT glacial values. I think the authors might be referring to one particularly large glacial maximum at ~1.5 Ma. Please clarify exactly what is referred to here and how it provides support for the new d18Osw record.

We have revised as:
This implies that site 1123 is recording large ice sheets at ~1.5 Ma, or before the MPT.

Lines 605-606: The same text is repeated on these two lines.

Thank you – now corrected.

Lines 647-648: The meaning of this sentence isn't clear. Ice sheets have enhanced the warming relative to what? How is this visible in Figure 14C?

Thank you. Clarified as:
The effect of lowering of Northern Hemisphere ice sheets as they retreat induces surface warming…

Figure 1: Many of the individual records are partially/mostly hidden behind other data in this figure. Also, the caption suggests that there are two different orange lines in the figure, which seems like a problem.

Now only one orange line. We don't think it's necessary to completely see every record – the main point is that there is a large spread in the reconstructions.

Figure 10B: It's very hard to see the light blue line (which is an important result to be able to see) due to overlap with the gray line. Maybe make the shade of blue darker or leave off the gray line.

We have darkened the blue line.

Figure 16F: The caption doesn't provide the color information for all the different records shown.

Now added.

---

## Author Comment (AC2)

This paper is a tour-de-force addressing difficult but highly relevant question of global climate evolution over the past 4.5 M years. This required the disentanglement of surface and deep ocean temperature under the constraint of global sea level rise that is derived from the benthic 18O stack. The analysis considered many other available high-resolution records covering this period and encompasses two apparently very different climatic regimes: the warm period before 1.5 Myr with smaller amplitude ice sheet and sea level cycles, primarily on the obliquity time scales, and after the Mid-Pleistocene Transition (MPT), i.e., post 1 M yr the familiar and well documented large-amplitude ice age cycles. The central result is that during the MPT the ocean heat storage efficiency (HSE) must have shifted from a low, constant value before, to twice that value, again constant, afterwards. This argument hinges on subtle differential changes between global mean SST and MOT that seem to be time dependent which leads to the hypothesis of a change in HSE.

Thank you for this positive assessment and very helpful comments which have improved the manuscript.

The strength of this paper is the compilation and the comprehensive discussion of the wide paleoclimatic evidence from the various high-resolution archives. The weakness, however, is the motivation for the significant change in HSE and the underpinning modelling framework. Irrespective of this, the present contribution is important and forms part of a series of papers that address the structure and dynamics of global-scale climate during the last 4.5 M years. The authors should be encouraged to revise the paper, clarify the points raised in this review and provide a firmer and more robust modelling basis for their important conclusions.

We're not quite sure what the reviewer means by the weakness of our study is the "motivation" for the significant change in HSE. If motivation means understanding the cause of the change, we first point out that, following the reviewer's suggestion below, we downloaded the 15 model results from the PLIOMIP2 experiment (e.g., Haywood et al., 2020, *Climate of the Past*) to obtain ΔGMSST and ΔMOT for each model run. We added these to the model results shown in Fig. 2A and reassessed the HSE using several statistical models. As explained in the revised text, two statistical models (LOESS and segmented regression with two breakpoints) provide equivalent fits to the data that are superior to the linear regression used in the original Fig. 2A. These model results now suggest that HSE is 1 for ΔGMSST <~0ºC, 0.6 for ΔGMSST ~0ºC to ~5ºC, and 1.2 for ΔGMSST >~5ºC. These results thus agree with what we derived from the proxy data for the last 4.5 Ma, providing robust evidence for an increase in HSE during the MPT from ~0.5 to 1. These new results have been incorporated into the revision.

Regarding the "underpinning modeling framework" discussed in section 4, the main purpose here is to use model output from cold and warm climate states to illustrate the primary mechanisms of ocean heat uptake to make the following point in the paper (we note that the newly added PlioMIP2 data show the patterns of ocean heat uptake as seen in our other examples):
"This understanding of the major processes involved in ocean heat uptake and storage in response to GHG emission scenarios over the course of this century (Fox-Kemper et al., 2021; Cheng et al., 2022) or on equilibrium timescales ($10^3$ yr) (Fig. 14A, 14B) (Rugenstein et al., 2016; Li et al., 2013) contrasts with the longstanding view in paleoceanography that changes in

DOT and MOT result solely from SST changes in high-latitude regions where deepwater is formed (Emiliani, 1954; Zachos et al., 2001; Hansen et al., 2023; Evans et al., 2024; Westerhold et al., 2020; Bereiter et al., 2018; Rohling et al., 2022; Hansen et al., 2013). Changes in source water temperature may indeed cause changes in deepwater temperature, for example during substantially warmer climates without Antarctic sea ice (Goudsmit-Harzevoort et al., 2023; Evans et al., 2024) when AABW could form during Antarctic winter solely by heat loss to the atmosphere without brine rejection, like NADW in the present climate. However, the general notion of a sole control of MOT by SSTs at sites of deepwater formation (affecting $T_d$) should not be applied regardless of climate state because it neglects the roles of the rate (v) of deepwater formation, as well as the contribution from ocean heat uptake in mid-latitudes, both of which can also affect ocean heat storage (Fig. 14A, 14B) and thus contribute to MOT."

We then use this understanding to develop our simple scaling analysis in section 5 which uses model parameters for different climate states (volume of two main heat reservoirs, relationship of upper heat reservoir to GMSST) to partition the heat between the two reservoirs under different climate states.

Major comments:
1) A key conclusion is the HSE changes by a factor of 2 crossing the MPT. Hence, fundamentally the MPT is seen as a change in the ocean-atmosphere system through a combination of changes in sea ice cover, i.e. the transfer of heat from the atmosphere to the ocean, and changes in the repartitioning of heat between the surface/upper ocean and the deep ocean.

We do interpret the cause of the MPT as a combination of these changes, but we view this combination more correctly as cause and effect, whereby the changes in sea ice cover contribute to (with decreasing temperature at source regions) and changes in ocean heat uptake through their effect on deepwater formation.

The argument is motivated by an earlier study using a global comprehensive, isotope-enabled climate model (iCESM1) that simulated the last deglaciation (Zhu et al., 2024).

The goal of the Zhu et al. (2024) study was to try to understand why data-constrained HSE during the last deglaciation remained ~1. Zhu et al. used single forcing experiments to identify the contribution of different forcings to HSE during the deglaciation. In this regard, the study was looking at **transient** changes in HSE. It found that without freshwater forcing, transient HSE was <<1, and it remained at ~1 throughout the deglaciation only with the addition of a shutdown of the AMOC, but in **all** forcing cases, **equilibrium** HSE was ~1. This study was thus not the motivation for our present study which is trying to understand the cause of the increase in **equilibrium** HSE from ~0.5 to ~1.

What the Zhu et al. study does provide for us here are examples of the primary mechanisms of ocean heat uptake. When compared to simulations with warmer climates (including now the PlioMIP2 results), we show that the mechanisms are robust under a range of climates from colder than to warmer than present, albeit with different magnitudes. The Zhu et al. study (and

now the PlioMIP2 study) also provide us with the parameters we need for our simple scaling analysis described in section 5.

It is not clear whether the different climate states that are visited in the 20 kyr-simulation encompass those that are relevant during the past 4.5 M years. I think the LGM-BA-YD-Hol sequence of the simulations is quite representative of the states that are visited after the MPT. However, I am not convinced that the same holds true for the climate states prior to 1.5 Myr ago. The reason is that generally higher global mean temperatures prevailed then with a quite different SST and polar temperatures. Overall, this was a situation of a substantially different ocean climate, particularly with respect to stratification which is the key process that may regulate HSE. The question is whether a comprehensive model under pre-MPT conditions would show a diminished HSE.

As explained above, the different climate states in the iTRACE simulations are relevant to this study by providing examples of the main processes of ocean heat uptake and to provide a range of parameters we use (for sensitivity) in our scaling analysis for the post-MPT HSE=1 climate state.

2) It appears that PlioMIP would be able to provide this underpinning. The model simulations presented in Weiffenbach et al 2024 are, unfortunately only analysing Southern Ocean processes, but surely the modelling results would be available to determine deltaSST, deltaMOT and all the quantities that are required to estimate HSE for that period. With such an analysis, all arguments in the present version to support a reduced HSE prior to the MPT could checked quantitatively. I am aware that this would be substantial additional work, but it would deliver the underpinning for the claims made in the paper.

This is an excellent suggestion and we have now downloaded the PlioMIP2 data. The HSE based on the mean of the 15 models is 0.78, which is thus consistent with our argument that it is <<1. We have included the individual model results as well as the multi-model mean on the revised Fig. 2 which reinforce the statistical argument that HSE was ~1 for GMSST <~0°C, ~0.5 for GMSST ~0°C to 5°C and ~1.2 for GMSST >5°C. We also use the multi-model mean parameters of the volume of the two heat reservoirs and the relationship of the temperature of the upper heat reservoir to GMSST for the >MPT interval in our scaling analysis.

3) Figures 14 and 15 should provide model-based insight for the HSE argument. However, it is not straightforward for the reader to connect the three cases (4x, mid Hol limited forcing, mid Hol full forcing) shown in Fig 14 with the conclusion of different HSE. Might the average of panel B be and approximation to HSE and (by visual estimate) about 1?? If so, the mean of panels D and F would also be approximately HSE with presumably HSE > 1??

The purpose of these figures is to show the mechanisms by which ocean heat uptake occurs, not to conclude that there is a different HSE. The key argument is that each simulation shows that ventilation of intermediate waters plays an important role under all climates (surface winds are always blowing) whereas deepwater formation is sensitive to surface boundary conditions as opposed to the conventional argument that it is all from deepwater formation (see above).

We have added the equivalent figures for the PlioMIP2 results and revised the figure caption to read:

**Figure 14.** Changes in 0-2000 m mean equilibrium temperature change (left) and zonal average equilibrium ocean warming pattern normalized with ΔMOT (right) for different modelling scenarios. **(A, B):** CESM 1.0.4 for abrupt $4xCO_2$ minus preindustrial after 5900 years, ΔMOT=5.12K (Rugenstein et al., 2019). **(C, D):** iTRACE ICE+ORB+GHG simulation for mid-Holocene (6 ka) minus LGM, ΔMOT=3.02K (Zhu et al., 2024). **(E, F):** full iTRACE simulation (ICE+ORB+GHG+AMOC) for mid-Holocene (6 ka) minus LGM, ΔMOT=3.6K (Zhu et al., 2024). **(G, H):** Model mean of PlioMIP2 simulations, ΔMOT=1.86K (Haywood et al., 2020).

Clarification of why you here show scaled ocean temperature changes would be appreciated.

Showing the temperature change normalized to average ocean temperature change is a standard way of identifying patterns that represent processes of ocean heat uptake (e.g., Cheng et al., 2022; Zhu et al., 2024).

The message of Fig 15 is not clear, and the figure could be possibly omitted.

Figure 15 provides information (enhanced surface warming in the North Pacific and North Atlantic) that helps understand the enhanced ocean heat uptake through ventilation of intermediate waters at 45°N seen in Figure 14D and so we prefer to keep it.

4) Further to the modelling, the %-changes stated on line 690 seem quite fundamental to the argument, yet they pop up as a surprise and their derivation is not clear. I also do not find such analysis in Zhu et al 2024. Furthermore, how do these numbers connect with HSE? Where is 1.16 coming from, a few lines down? It seems that section 5 needs a major revision to make a convincing, model-based case.

As stated in the text, these come from the iTRACE modeling results in Zhu et al. They were not used (or mentioned) in that paper, but we extracted them from those model results for our use here. As explained in the text, these establish the average temperature change of the upper reservoir $\Delta T_{<2000}$ relative to ΔGMSST, which we then use to calculate the temperature of the deep ocean after also accounting for the relative ocean volumes of the two reservoirs. The 1.16 value is stating that the temperature of the upper reservoir is 16% greater than GMSST (i.e., one of the % changes listed on line 690). We have clarified this as:

"From the iTRACE results (Zhu et al., 2024) we calculate that the average temperature change of the upper reservoir $\Delta T_{<2000}$ is 11%, 16%, and 54% greater than ΔGMSST for scenario ICE+ORB+GHG, the full scenario that includes meltwater forcing, and the ICE-only scenario, respectively. This corresponds to a multiplying factor $s=\Delta T_{<2000}/\Delta GMSST$ of 1.11, 1.16 and 1.54, respectively."

This information is introducing the main parameters and terms (all model based) used in our scaling analysis, which we think is adequate to physically understand the main changes that may have caused the increase in HSE across the MPT. We don't see the need for a major revision.

Further comments:

5) Fig. 1, Caption: various shifts are mentioned. The shift by -1.73°C is motivated twice, first owing the entire 800 kyr, then to the Holocene. This is confusing.

In the two places where shifts are mentioned, we clarified as follows:
Mg/Ca BWT's from this data set were mean shifted…

6) Footnote 2: HSE is referring to equilibrium, HUE to transient changes. An important paper of reference is Zhu et al. 2024. In their figure 4, HSE is given as a function of time throughout the last 20 kyrs, in fact with longer periods where HSE>1, a case that is never discussed in this paper. A clarification of the different concepts and the relation to Zhu (used many times later in the paper) would be important.

The purpose of this footnote is to clearly distinguish (and avoid confusion) between HSE, which is used here in the same way defined in Zhu et al. (2024) (i.e., **equilibrium** change) and ocean heat uptake efficiency (HUE, mentioned by the reviewer), which (as explained in the footnote) describes **transient climate states on decadal timescales** and is commonly used when looking at recent changes in ocean heat uptake (e.g., Newsom et al., 2023, GRL. https://doi.org/10.1029/2023GL105673). The footnote is referenced to the sentence in the main text that refers to Zhu et al. (2024) as the source for the definition of HSE, which we then use throughout the paper to refer to ΔMOT/ΔGMSST. Since HSE is ~1 when in equilibrium, Zhu et al. (2024) evaluated why it should be ~1 during the transient deglaciation, as suggested by the data, but this is irrelevant to our study.

7) Fig 2B: x-spread of red dots (LGM temperatures) is significantly less than purple dots (all temperatures of last 07. Myr) and yet the correlation is the same and the regression line goes far beyond the rightmost red dot (as far as I can recognize enlarging the figure)- I would expect a much reduced r^2 for the red data points.

There are the same number of red and blue dots – where you only see blue dots means that the red dots have the same value (i.e., they are behind the blue dots). This is why the red regression line continues to the upper right corner. As explained in the caption, the red dots that you can see (i.e., that are not the same as the blue dots) are glacial-age data to which we added 0.38°C. This is just a simple sensitivity test of our results when accounting for the potential cold bias in glacial MOTs discussed by Seltzer et al. (2024).

8) Introduction: the intro is rather short, essentially only lines 43 to 62. From 63 to 121 a description of the current paper is given. Line 81, continued at 105 already provides the conclusion without having first given the overall context for HSE.

We're not sure how to respond to this other than the statement that "Line 81, continued at 105 already provides the conclusion without having first given the overall context for HSE" is inaccurate. This text introduces HSE as a simple parameter for representing ΔMOT/ΔGMSST, with the main point of this text setting up the main issue of the paper, i.e., why did HSE increase across the MPT?

9) Fig 4 I: different time axis. Not sure whether this may be a mistake. Use the same time axis as in other panels; if there is no data prior to 3.4 Myr, then this should be left blank.

This is no mistake. The point of the figure is to compare our GMSST reconstruction to data, so in this case, we want to emphasize that comparison by keeping the time axis for Fig. 4I as is. Also note that in the comparison we show in Fig. 3, the different panels also have different time axes.

10) Fig 4 C: is there a cutoff in the red data at -2°C, but not in others?

This is as the data have been published originally. What looks like a cutoff is a lower limit for temperature change the original authors assumed to keep their time series above the freezing point of seawater.

11) entire paper: deltaMOT and deltaGMOT – what is the difference?

This was an oversight. It should just be (and now is) ΔMOT.

12) Fig 5B: refer again explicitly to Clark et al 2024 for the black line.

Done.

13) line 309: specify what long-term means.

We have revised this as "so that they are the same as the long-term (401-kyr running average) mean as our ΔMOT reconstruction."

14) line 317: this assertion is presumably based on visual inspection. If some quantitative approach is used, please specify.

We have revised as "We find that the best agreement with the Mg/Ca-based ΔBWT data based on their similar slopes…"

15) line 324: clarify that the grey curve and cloud is based on HSE = 1 throughout the entire 4.5 Myr.

Done.

16) Fig 7: add labels of region to the panel, e.g. for A "eq Pacific".

Rather than clutter the figures, this information is in the caption.

17) Fig 8 could be made more compelling by shifting downward the HSE curve so that it does not overlap with the two time series. Ad separate y-axis on the rhs, occupying much less vertical space for such a trivial curve (two straight lines linked by a slope).

We have shifted the HSE curve downwards, but although this is a "trivial" curve, it nevertheless explicitly shows how we have defined HSE.

18) Fig 9: the Mg/Ca constraint is really only operating in 1.5 to 3.2 Myr. This should be emphasized.

No – the Mg/Ca data shown include those from Lear et al. (2003) which extend to 4.5 Ma.

19) line 384: should this be 0 to +0.1 permil?

Yes – changed.

20) Fig 9, line 371: in text you use secular, here long-term. Please make consistent (btw secular would strictly mean century-scale (Latin saeculum), but likely you mean million-scale?)

We have removed the term "secular" from the text altogether.

21) line 449: should this rather be "increase" in d18O_T?
Yes – thank you. Changed.

22) Fig 13: it would be helpful to have the ODP labels and locations at the right of the panel rows.

Again, we prefer not to clutter the figure when this information is easily available from the caption.

23) line 553: salinity changes are mentioned which is interesting. Could this be quantified. What would be the required magnitude? It this reasonable?

This cannot be quantified.

24) line 580: perhaps you add 30S to 30N and upper 500 m for the majority of heat uptake so the information follows the one on line 579.

We state that the majority of ocean heat uptake occurs within wind-driven subduction regions in the Southern Ocean.

25) line 583: Pacific warming is also smaller due to upwelling of colder intermediate and deep water.

We're talking about area-averaged warming for the Pacific as a whole, not that part that might be influenced by upwelling.

26) Fig 14 line 599: not clear which iTRACE simulations are meant. Once you use "full simulation" once just "simulation".

Thank you – now clarified as "full"

27) line 605: duplication of sentence.

This was an oversight – now fixed.

28) line 688: this is a very simple repartitioning model. It may serve the purpose, but the f's and the deltaT's should be diagnosed from model simulations. It is currently a bit obscure how these values (e.g., line 700, f=0.43) come about.

The f's are diagnosed from model simulations, as in "the upper ocean reservoir…has a volume that, in iTRACE, is 43% of the global ocean (f<2000 = 0.43)." $\Delta T$ for the upper reservoir is based on its temperature being some percentage (i.e., 11%, 16%, 54%) greater than GMSST, also based on the model. $\Delta T$ of the lower reservoir is then just $\Delta MOT$ minus $\Delta T$ of the upper reservoir.

---

## Author Response (AR2)

March 4, 2025

Denis-Didier Rousseau
Co-editor-in-chief
*Climate of the Past*

Dear Denis,

Please find below our responses (in red) to the four comments by Lorraine Lisiecki, where we have revised the paper in each case according to her suggestions. I hope you find these satisfactory.

Thank you for your very efficient handling of this paper.

Best regards,

Peter

- Line 36: I found this sentence difficult to parse. Consider "These results point to changes in deep ocean circulation … as the cause of increased ocean heat uptake and HSE."

We changed this to read:
These results point to an increase in ocean heat uptake and HSE during the MPT in response to changes in deep ocean circulation driven largely by surface forcing of the Southern Ocean.

- Lines 678-680: I have trouble understanding how this sentence follows from the previous one. Please clarify whether the warming at 45N is causing the Southern Ocean warming or if these are associated with two separate mechanisms.

We changed this to read:
    Ice-sheet forcing in the ORB+ICE+GHG simulation thus significantly enhances intermediate-depth warming that is otherwise largely occurring through wind-driven ventilation in the Southern Ocean (Fig. 14C).

- Line 692-694: Is this lag estimate derived from the ICE+ORB+GHG simulation? If so it might be worth clarifying to draw a contrast with the next paragraph that includes meltwater forcing.

Yes. We have clarified in two places that the lag is found in the ICE+ORB+GHG simulation:
    During the deglaciation, the iTRACE ORB+ICE+GHG simulation shows that MOT warming lags SST warming by several thousand years, reflecting the timescale of warming the ocean interior by ventilation of intermediate waters and resulting in an average transient HSE ~0.5 that disagrees with proxy records (Zhu et al., 2024). However, this simulation shows that the lag is too short to prevent equilibrium MOT warming from being reached during the present interglaciation and HSE reaches ~1.

- Line 695: I found it hard to parse the phrase "the lag is too short to prevent equilibrium MOT

warming". Why not state it more directly as something like "the lag is short enough to allow MOT warming to reach equilibrium…"

We have changed this as suggested.